# An Efficient Certificateless Aggregate Signature Scheme for Blockchain-Based Medical Cyber Physical Systems

**DOI:** 10.3390/s20051521

**Published:** 2020-03-10

**Authors:** Hong Shu, Ping Qi, Yongqing Huang, Fulong Chen, Dong Xie, Liping Sun

**Affiliations:** 1School of Mathematics and Computer, Tongling University, Tongling 244061, China; shuhongtl@126.com (H.S.); qiping929@gmail.com (P.Q.); hyq@tlu.edu.cn (Y.H.); 2Anhui Provincial Key Lab of Network and Information Security, Wuhu 241002, China; xiedong@ahnu.edu.cn (D.X.); slp620@163.com (L.S.); 3Institute of Information Technology & Engineering Management, Tongling University, Tongling 244061, China; 4School of Computer and Information, Anhui Normal University, Wuhu 241002, China

**Keywords:** blockchain, privacy protection, certificateless aggregate signature, trapdoor hash function, MCPS

## Abstract

Different from the traditional healthcare field, Medical Cyber Physical Systems (MCPS) rely more on wireless wearable devices and medical applications to provide better medical services. The secure storage and sharing of medical data are facing great challenges. Blockchain technology with decentralization, security, credibility and tamper-proof is an effective way to solve this problem. However, capacity limitation is one of the main reasons affecting the improvement of blockchain performance. Certificateless aggregation signature schemes can greatly tackle the difficulty of blockchain expansion. In this paper, we describe a two-layer system model in which medical records are stored off-blockchain and shared on-blockchain. Furthermore, a multi-trapdoor hash function is proposed. Based on the proposed multi-trapdoor hash function, we present a certificateless aggregate signature scheme for blockchain-based MCPS. The purpose is to realize the authentication of related medical staffs, medical equipment, and medical apps, ensure the integrity of medical records, and support the secure storage and sharing of medical information. The proposed scheme is highly computationally efficient because it does not use bilinear maps and exponential operations. Many certificateless aggregate signature schemes without bilinear maps in Internet of things (IoT) have been proposed in recent years, but they are not applied to the medical field, and they do not consider the security requirements of medical data. The proposed scheme in this paper has high computing and storage efficiency, while meeting the security requirements in MCPS.

## 1. Introduction

In the big data era, with the development of Internet of Things, smart healthcare provides people with more convenient and high-quality healthcare services [1]. The Medical Cyber Physical System (MCPS) [2] is a special type of Cyber Physical System (CPS) based on the application background of the smart healthcare field, which consists of physical space and cyber space. Physical space includes wearable devices, medical diagnostic equipment, and user space consisting of doctors, nurses, etc. Cyber space is the nerve center of MCPS. It receives sensing information from physical space through a network transmission system. Then cyber space identifies, stores, analyzes, processes, and generates feedback control information. Finally, it sends control information to physical space through a network transmission system.

MCPS continuously collects the patient’s physical signs data through various wearable devices and medical devices, so that the patient’s physical condition can be better detected [3]. In order to provide patients with a more accurate and timely diagnosis, different medical institutions need to share a large amount of physical data collected by the sensors and healthcare staff [4]. At the same time, patient privacy should be protected. Thus, blockchain is needed to utilize peer-to-peer network and cryptography technology to achieve tamper proof, unforgeable, non-repudiation, and verifiable medical records. The combination of MCPS and blockchain [5] promotes the sharing of medical services and resources [6]. However, the block capacity limit is one of the main factors that affects the performance improvement of blockchain.

MCPS controls the embedded medical equipment through a wireless network, which senses and monitors the patient’s physical data in real time. When the patient has an abnormal situation, the medical equipment sends the early warning information to the medical institution in time. Once MCPS is under cyberattacks, such as data inconsistency, unauthorized access, and data breaches [7], patients’ lives and health will be seriously threatened. In practice, medical institutions need to check the accuracy and integrity of shared and sensed medical data before making medical diagnoses. The medical data, which is collected from wearable devices, medical equipment, medical apps, and healthcare staff needs the responsible healthcare provider to sign on it. A large number of signatures and verifications result in high time and space overheads. At the same time, considering the capacity limitation of the blockchain, the certificateless aggregate signature is an effective method because of its compression characteristics. In recent years, some certificateless aggregate signature schemes [8,9,10] have been proposed. However, the performance of these schemes is not ideal because they use more time-consuming bilinear maps. At the same security level, the Elliptic Curve Cryptography (ECC) is more efficient than bilinear maps [11]. Therefore, with the characteristics of low computation, low storage, high reliability, privacy protection, and timeliness, the certificateless aggregate signature scheme based on ECC is suitable for blockchain-based MCPS.

The contributions of this paper are as follows:A two-layer storage model in which medical data is stored off-blockchain and shared on-blockchain is proposed. The model meets security and privacy requirements of MCPS.Based on ECC, we present the multi-trapdoor hash function, which is secure and efficient to construct the certificateless aggregate signature scheme.The certificateless aggregate signature scheme based on the multi-trapdoor hash function is proposed in this paper. It can reduce the computation cost of wearable medical devices and miners.

The rest of this paper is organized as follows. Related works are discussed in Section 2. The necessary preliminaries are presented in Section 3. Section 4 presents a multi-trapdoor hash function. In Section 5, we describe the certificateless aggregate signature scheme. A security discussion of the proposed scheme is given in Section 6. Then, we make an efficiency analysis in Section 7. Finally, the conclusion is offered in Section 8.

## 2. Related Work

### 2.1. Blockchain

Blockchain is a decentralized, anonymous, untrusted, tamper proof, and traceable distributed data storage technology [5]. With the development of the medical industry, health data is growing exponentially. How to effectively store, share, and manage medical data involving a large number of patients’ privacy has become an obstacle to the development of the healthcare industry. Due to the characteristics of blockchain [12], such as non-tamperability, traceability, and multi private key authorization management, it is possible to share medical data securely among different institutions [13]. According to the difference of open objects, blockchain can be divided into Public Blockchain, Private Blockchain, and Consortium Blockchain. These three types of blockchains are compared in Table 1. In the special field of MCPS, medical data contains both a large amount of private information and has the need to be shared between different institutions, therefore the Consortium Blockchain is more suitable for the secure storage and sharing of medical data.

Xue et al. [14] divided the existing medical institutions into medical institution federate servers (MIPS) and audit federate servers (AFS) according to their credit scores. Through the improved consensus mechanism, the medical data sharing model based on blockchain was realized. In the untrusted environment, Xia et al. [15] designed a sensitive medical data sharing model between cloud service providers based on blockchain through a smart contract and access control mechanism. The security requirements of medical records on integrity, confidentiality, and traceability can be realized by digital signature technology in the blockchain-based medical data sharing system.

In recent years, researchers have conducted in-depth research around blockchain-based multi-signatures [16], aggregate signatures [17,18], ring signatures [19], and homomorphic signatures [20]. Among them, aggregate signatures are favored for their advantages, such as fast computing speed, small storage space, and bandwidth saving. Moreover, some scholars have carried out in-depth research on the combination of quantum computing and the security of blockchain [21]. Gao et al. [21] proposed a lattice-based signature scheme and presented a cryptocurrency scheme based on post-quantum blockchain, which could resist quantum computing attacks.

### 2.2. Certificateless Aggregate Signature

In order to solve the management problems of certificate distribution and storage in the traditional PKI-based (Public Key Infrastructure) public key cryptosystem, Shamir proposed the identity-based public key cryptosystem (ID-PKC) in 1984 [22]. In ID-PKC, the public key is denoted by user information, such as mailbox, address, telephone number, etc. The private key is provided by the key generation center (KGC), a third-party trusted organization. Different from traditional public key cryptosystems, users cannot generate their own private key. For KGC, the user’s private key is known, and KGC can decrypt ciphertext and forge identity at will. Therefore, ID-PKC has the defect of key escrow [23], which is only applicable to the environment with low security requirements.

To solve this problem, Al-Riyami and Paterson proposed the notion of certificateless public key cryptography (CL-PKC) in 2003 [24]. Unlike ID-PKC, the private key in CL-PKC consists of a partial private key generated by KGC and the secret value selected by the user. KGC only knows partial private key but cannot get the secret key. It can effectively solve the key escrow problem [25]. Moreover, the public key in CL-PKC does not need certificate verification, so the problem of public key authentication is solved. CL-PKC has neither the certificate management problem nor the key escrow problem. Its calculation efficiency is higher than traditional public key cryptosystems, and its security is higher than ID-PKC. Therefore, it is suitable for application scenarios with higher requirements for computing, storage efficiency, and security.

Boneh et al. first proposed the concept of aggregate signature [26] on EUROCRYPT 2003, which greatly promoted the development of digital signature cryptography. Aggregate signature [26] is suitable for compressing many signatures generated by many different users to many different messages into one short signature, and simplifying the verification of multiple signatures into one verification. Aggregation signature greatly improves storage efficiency and verification time.

In recent years, certificateless aggregate signatures (CLAS) have attracted many scholars’ research interests because of the advantages of both a certificateless public key cryptosystem and aggregate signatures. Based on different theoretical foundations, scholars have proposed corresponding certificateless aggregate signature schemes. For example, most researchers proposed certificateless aggregate signature schemes based on bilinear maps [8,9,10]. For the first time, Gong et al. [9] proposed two certificateless identity-based aggregate signature schemes (denoted as CAS-1 and CAS-2 in [9]). In these two schemes, the aggregation verification of CAS-1 used 2*n* + 1 pairing operations on an elliptic curve. CAS-2 used *n* + 2 pairing operations and *n* scalar point multiplication operations on elliptic curves. It is clear that the verification efficiency was very low. Xiong et al. designed a more efficient certificateless aggregate signature scheme [8]. The verification of this scheme used only three pairing operations and 2*n* scalar multiplication operations. The efficiency of the scheme was not related to the number of signers. Moreover, it did not require a synchronized clock. As such, this scheme was more efficient than the Gong’s scheme [9]. However, He et al. [27] and Zhang [10] et al. pointed out that Xiong et al.’s scheme was not secure. He et al. [27] proved that Xiong et al.’s scheme was not resistant to forge attacks from **A****_Ⅱ_** adversary. Zhang et al. proved that Xiong et al.’s scheme could not resist coalition attacks from the honest-but-curious KGC, malicious-but-passive KGC, and inside signers.

Some scholars did not use bilinear pairs to construct certificateless aggregate signatures. Zhou et al. proposed two certificateless aggregate signature schemes without bilinear maps [28]. Based on the Elliptic Curve Discrete Logarithm Problem (ECDLP), the schemes both used 2*n* + 1 scalar multiplication operations. The difference is that CLAS-2 provides a shorter constant-level signature length than CLAS-1. Cui et al. [29] proposed a certificateless aggregate signature scheme based on ECC and applied it to vehicular ad hoc networks (VANETs) communication. The verification of this scheme used *n* scalar multiplications. Since the computational overhead of bilinear pairs is significantly higher than that of scalar multiplication under ECC [11], Zhou’s scheme and Cui’s scheme had higher computational efficiency.

In recent years, with the development of blockchain technology, more and more scholars have focused on the research of the aggregation signature algorithm based on blockchain [17,18,30]. Gao et al. [18] designed a fair and efficient multi-party contract signing scheme based on blockchain by conducting a certificateless aggregation verifiable encryption signature scheme. Wang et al. [30] realized the full anonymous blockchain by homomorphic encryption, and aggregate signature technology, which effectively protected the privacy of the user’s identity and the transaction amount. Neither of these schemes [18,30] is computationally efficient because they both used bilinear maps. Based on the gamma signature proposed by Yao et al. [31], Zhao [17] constructed an aggregate signature scheme without bilinear maps. By applying Zhao’s scheme [17] to Bitcoin, it could be found that both computation and storage overhead have decreased to some extent, however the length of this aggregate signature scheme increased with the number of signers. Due to their low computing or communication efficiency, these schemes [17,18,30] were not suitable for wearable medical devices with limited computing and storage resources. On the other hand, these schemes [17,18,30] did not focus on the security requirements of MCPS, such as timeliness and privacy protection.

Some scholars focused on the research of digital signatures in blockchain-based Internet of things (IoT) applications [32,33]. In order to reduce the time cost of transmitting authentication information from blockchain nodes to IoT devices, Danzi et al. [32] proposed a repeat-authenticate scheme. In which blockchain information that consists of a copy of the block header and the signatures of blockchain nodes is multicasted. Kaga et al. [33] proposed a biometrics-based fuzzy signature scheme and applied it into the IoT blockchain system. This scheme achieved the verification of a creator of a transaction. These two schemes payed more attention to authentication of transaction creators or blocks in IoT scenario. However, they did not focus on the effective storage of a large number of digital signatures and the privacy protection of medical data in MCPS scenario. When a patient goes to the hospital, a great deal of medical records will be generated. The digital signatures of these medical records will occupy a large amount of block space, which will seriously affect the performance of the blockchain. At the same time, medical data involves personal privacy, and it is necessary to protect the private data. 

The blockchain-based schemes mentioned above are compared in Table 2. From Table 2, we can conclude that none of these solutions [17,18,30,32,33] provide both high computing and communication efficiency. Furthermore, nowadays, certificateless aggregate signatures based on blockchain have not been widely used in MCPS. In this paper, we combine ECC and the multi-trapdoor hash function to propose a certificateless aggregate signature scheme and apply it to secure storage and sharing of MCPS. The proposed scheme provides high computing efficiency and low space occupation, which is suitable for blockchain-based MCPS scenario with limited blockchain capacity and low computing power wearable devices.

## 3. Preliminaries

### 3.1. Elliptic Curve Discrete Logarithm

Let p, q be two large prime numbers, F_p_ be a finite field determined by p, and E(F_p_) be an elliptic curve over F_p_, which is defined by the equation: *y^2^ = x^3^ + ax + b* mod p, where a, b∈F_p_ and 4a^3^ + 27b^2^ ≠ 0. If the additive group G consists of the infinity point O and all points on E(F_p_), P is a generator of group G with the order q, we have the following definition.

**Definition** **1**(Elliptic curve discrete logarithm problem (ECDLP) [34])**.**
*Given Q = mP and*
$\;Q\in {\;E(F}_{p})$*, the task of ECDLP is to find the integer m, where 0 ≤ m ≤ q − 1.*

### 3.2. Trapdoor Hash Function

The trapdoor hash function is also called the chameleon function [35]. Different from general hash functions, it has a hash/trapdoor key (HaK, TrK). The hash key (HaK) is public, while the trapdoor key (TrK) is private. The trapdoor hash function uses some special information to generate a fixed hash value, and its collision resistance depends on the user’s knowledge of trapdoor information (TrK) [36]. That is, without knowing the trapdoor key TrK, the trapdoor hash function is collision resistant. However, when the hash/trapdoor key is known, the trapdoor collision can be computed [37]. This property of the trapdoor hash function is suitable to construct various digital signature schemes [36,37,38,39].

The trapdoor hash function consists of the following four algorithms [37]:
**ParG**: Inputs security parameter *k*, outputs system parameter *params*;**KeyG**: Inputs *params*, outputs hash/trapdoor key <*HaK*, *TrK*>;**HashG**: Inputs *params*, message *m* and auxiliary parameter *r*, outputs trapdoor value *TH_HaK_*(*m*, *r*);**TrapColG**: Inputs *params*, <*HaK*, *TrK*>, *m*, *r,* and new message *m*’(≠ *m*), outputs *r*’ and *HaK*’ such that ${TH}_{HaK}\left(m,\;r\right){\;=\;TH}_{{HaK}^{\prime }}{(m}^{\prime }{,\;r}^{\prime })$;

According to the number of trapdoor information (*TrK*), trapdoor hash functions include the single trapdoor hash function [35], the double trapdoor hash function [39], and the multi-trapdoor hash function [37,38]. A double trapdoor hash function usually has two pairs of hash/trapdoor keys, named long-term hash/trapdoor key and temporary hash/trapdoor key. Double trapdoor hash function protects the long-term trapdoor key from being leaked by sacrificing the temporary trapdoor key. The multi-trapdoor hash function has multiple hash/trapdoor keys, which combines multiple collisions generated by multiple entities to conduct a single collision. As a result, the multi-trapdoor hash function has the advantage of computing efficiency as well as storage space and bandwidth saving. In this paper, we build a certificateless aggregate signature scheme based on the multi-trapdoor hash function, with which a blockchain- based MCPS data storage and sharing model is proposed.

### 3.3. Certificateless Aggregate Signature

#### 3.3.1. Definition of Certificateless Aggregate Signature

A certificateless aggregate signature consists of the following six algorithms [40]:
**Setup**: Inputs the security parameter *k*, KGC outputs the system public parameter *K_pub_* and system master key *λ*.**Partial-Private-Key-Gen**: Inputs *k*, *K_pub_*, *λ,* and user’s identity *ID_i_*, KGC outputs the partial private key *θ_i_* and sends it to the user *ID_i_* through a secure channel.**User-Key-Gen**: Inputs *k*, the user *ID_i_* outputs secret/public key pair (*α_i_*, *X_i_*).**Sign**: Inputs *k*, *ID_i_*, (*α_i_*, *X_i_*), and message *m_i_*, the user *ID_i_* outputs a signature *σ_i_*.**Agg-Sign-Gen**: Inputs *k*, ${{\{\;ID}_{i}\;\}}_{i\;=\;1}^{n}$, ${{\{\;\sigma }_{i}\;\}}_{i\;=\;1}^{n}$, the aggregator outputs the aggregate signature σ on ${{\{\;m}_{i}\;\}}_{i\;=\;1}^{n}$.**Agg-Ver-Gen**: Inputs *k*, ${{\{\;ID}_{i}\;\}}_{i\;=\;1}^{n}$, σ, ${\;\{\;m}_{i}{\;\}}_{i\;=\;1}^{n}$, and public key sets ${{\{\;X}_{i}\;\}}_{i\;=\;1}^{n}$, if the verification is correct, the verifier outputs 1, otherwise, the verifier outputs 0.

#### 3.3.2. Security Models of Certificateless Aggregate Signature

According to different capabilities, two types of adversaries are considered in certificateless aggregate signature schemes [9]. In addition, certificateless aggregate signature schemes should be existentially unforgeable under these adversaries, **A****_Ⅰ_** and **A****_Ⅱ_**.

**A****_Ⅰ_** adversary cannot get the system master key, but they can replace the public keys of legitimate users. Usually, **A****_Ⅰ_** adversary acts as malicious KGC.

**A****_Ⅱ_** adversary can obtain the system master key, however they cannot replace the public keys of legitimate users. **A****_Ⅱ_** adversary is often regarded as malicious inside signers.

For these types of adversaries, we define the following two games:

(1) Game Ⅰ:

**Setup**: Challenger Z inputs security parameters *k*, generates system parameter *pars* and system master key λ, sends *pars* to adversary **A****_Ⅰ_**, and keeps λ secretly.

**Query**: **A****_Ⅰ_** adaptively performs the following oracle queries:
*Hash queries*: **A****_Ⅰ_** sends a hash oracle query for all hash values in the scheme, and challenger Z returns the corresponding value.*Partial-Key-Gen query*: When **A****_Ⅰ_** makes a partial private key query on the user *ID_i_*, the challenger Z runs the partial private key generation algorithm to generate the corresponding partial private key ${\;\theta }_{i\;}$ and returns it to **A****_Ⅰ_****.***Secret-Key-Gen query*: When **A****_Ⅰ_** makes a secret key query on the user *ID_i_*, the challenger Z runs the secret key generation algorithm to generate the corresponding secret key $\alpha_{i}$ and returns it to **A****_Ⅰ_****.***Public-Key-Gen query*: When **A****_Ⅰ_** makes a public key query on the user *ID_i_*, the challenger Z runs the public key generation algorithm to generate the corresponding public key ${(\;X}_{i\;}{,\;V}_{i}\;)$ and returns it to **A****_Ⅰ_****.***Public-Key-Replacement query*: When **A****_Ⅰ_** queries user *ID_i_* for public key replacement, Z replaces the corresponding public key of user *ID_i_* with a randomly selected ${PK}_{{DAU}_{i}}^{*}{\;=\;(\;X}_{i}^{*}{,\;V}_{i}^{*}\;)$ and saves it.Signature queries: Inputs message $s_{i}^{\prime }$, user *ID_i_* and corresponding private key ${(\alpha }_{i}{,\;\theta }_{i}\;)$ and status information $\Omega_{i\;}$, Z runs the signature algorithm to generate the corresponding signature $\sigma_{i}$ and returns it to **A****_Ⅰ_****.**

**Forge**: After the above polynomial bounded queries, Z outputs the forged aggregate signature *σ^*^* = (*ω^*^*, *D^*^*). The adversary wins the game if and only if:
Forged signature *σ^*^* is a valid signature.**A****_Ⅰ_** cannot query at least one of *n* users for partial private key.

(2) Game Ⅱ:

**Setup**: Challenger Z inputs security parameters *k*, generates system parameter *pars* and system master key λ, sends *pars* and λ to adversary **A****_Ⅱ_**.

**Query**: In this stage, adversary **A****_Ⅱ_** adaptively performs the polynomial bounded oracle queries which are similar to Game Ⅰ. The difference is that **A****_Ⅱ_** does not perform the public key replacement query and partial private key query.

**Forge**: Z outputs the forged aggregate signature *σ^*^* = (*ω^*^*, *D^*^*). The adversary **A****_Ⅱ_** wins the game if and only if:
Forged signature *σ^*^* is a valid signature.**A****_Ⅱ_** cannot query at least one of *n* users for secret value.

### 3.4. System Model

In this paper, a two-layer system model is used to describe the secure storage and sharing of medical records in MCPS. As shown in Figure 1, the off-blockchain layer completes the acquisition, aggregation, and storage of medical data. In our proposed system model, every doctor, nurse, medical device, and medical app has a pseudonym, partial private key, secret value, and public key. The pseudonym is distributed by the Registry Center, and partial private keys are allocated by the KGC. Doctors, nurses, medical equipment, and medical apps are noted as data acquisition units (DAU). The medical record of a patient consists of several medical record items (MRI). Each MRI is signed by the DAU who is responsible for it. A patient’s diagnosis and treatment process corresponds to a Central Hospital. When a patient goes to different Central Hospitals, it corresponds to different treatment processes. Each DAU encrypts the collected MRIs with the public key of the Central Hospital, and calculates the hash value of MRIs it is responsible for as digital digest. The DAU’s private key is used to individually sign on the digest information. Then, the encrypted MRIs, digest information, and individual signatures are sent to the Central Hospital. The Central Hospital verifies the correctness of the individual signature. If it is correct, the encrypted original medical data is stored in the Medical Cloud. Finally, the Central Hospital combines the individual signatures into an aggregate signature, and sends the digest, aggregate signature, access control, and location index of the original MRIs to the Medical Blockchain.

The on-blockchain layer completes the sharing of medical data. Figure 2 shows that each transaction of the Medical Chain contains a digest of the *P_i_*’s MRIs, an aggregate signature, access control, and a specific location index of the original medical data stored in the Medical Cloud. Each block contains a hash value linked to the previous block. This hash value can be used to retrieve the block. The Medical Chain uses time stamps to ensure that the blocks are linked in time. The latest generated blocks are broadcast to the entire network. The nodes receiving the information verify the correctness according to the consensus algorithm. If it is correct, they pass the information to other nodes. After most nodes verify the correctness, the miner adds the block into the main chain to form the permanent storage and sharing of medical records. The patient is the owner of medical data, who grants an entity (doctor, institution, researcher, etc.) access to original medical records through access control protocol. When an entity gains access, they look up on the Medical Chain, obtains the position index of medical data in cloud, then they can access the original medical records.

In the above model, one block contains multiple transactions, and one transaction relates to all medical records of one medical treatment process of a patient. By using blockchain to store the digest and aggregate signature, the unforgeability of DAU’s service and the integrity of medical data can be guaranteed. Meanwhile, the block capacity limitation can be greatly eased. On the other hand, the encrypted original medical data is stored in the cloud, which is retrieved through the data location index on the blockchain. The access rights of entities are managed through the access control on the blockchain. Therefore, the secure storage and sharing of medical data in MCPS is realized.

### 3.5. Security Requirements

The following security requirements are important for medical data in MCPS:
Non-repudiation: Medical data is the record of treatment process, which has the function of legal evidence. Any modification of a medical record should be non-repudiation;Integrity: As an important record of the patient’s treatment, medical data should be guaranteed to be accurate, which means it cannot be tampered by anyone in any way. In other words, any data tampering can be detected;Privacy: Medical data involves patient’s personal privacy, which should be kept confidential. It could not be allowed to be disclosed at will, only the authorized users can access it;Traceability: When medical disputes occur between doctors and patients, medical data should be traceable as legal evidence;Timeliness: Time factor is one of the key points in the whole treatment process. It is necessary to make effective time judgment on each sensitive link in the treatment process, so as to ensure the authenticity and effectiveness of medical data.

Among these security requirements, tamper-proofing, data integrity, and privacy protection are crucial issues in MCPS [4]. It is necessary to use relevant technical means, such as identity authentication, blockchain technology, digital signatures, to achieve secure storage and sharing of medical information.

### 3.6. System Framework

The certificateless aggregate signature scheme based on the trapdoor hash function proposed in this paper consists of the following algorithms:
Setup: The algorithm is completed by KGC. Inputs security parameter *k*, outputs master key *λ*, system parameter *pars*.Pseudonym-Gen: The algorithm generates pseudonyms for each entity by Registry Center. Inputs the real identity of each *DAU_i_* or patient *P_j_* (denoted as ${RID}_{{DAU}_{i}}$ and ${RID}_{P_{j}}$), outputs its pseudonym ${PID}_{{DAU}_{i}}$ or ${PID}_{P_{j}}$.*DAU_i_* Key-Gen: *DAU_i_* generates its secret value-public key pair ${(\alpha }_{i}{,\;X}_{i})$ and sends ${\;X}_{i}$ to KGC through the secure channel. After receiving *DAU_i_*’s pseudonym ${RID}_{{DAU}_{i}}$, system parameters *pars*, public key ${\;X}_{i}$ and master key *λ*, KGC outputs the *DAU_i_*’s partial private key $\theta_{i}$. The public key (long-term hash key) of the *DAU_i_* is $X_{i}$, the long-term trapdoor key is $\alpha_{i}$, and the private key is $\theta_{i}$.Hash-Gen: In this algorithm, the trapdoor hash value of *DAU_i_* is generated. Inputs system parameter *pars*, original message *s_i_*, *DAU_i_*’s hash key $X_{i}$, auxiliary parameter *u_i_*, outputs *DAU_i_*’s trapdoor hash value ${TH}_{X_{i}}{(\;s}_{i}{,\;u}_{i})$.Individual-Sign: In this algorithm, *DAU_i_* generates its individual signature. Inputs system parameter *pars*, digest of MRIs in the charge of *DAU_i_* (denoted as $s_{i}^{\prime }$), status information *Ω_i_* of *DAU_i_*, trapdoor key $\alpha_{i}$, hash key $X_{i}$, and outputs *DAU_i_*’s individual signature $\sigma_{i}$.Individual-Verify: The Central Hospital verifies the correctness of individual signature. Inputs *DAU_i_*’s individual signature $\sigma_{i}$, hash key $X_{i}$, check the correctness of $\sigma_{i}$. If correct, accepts $\sigma_{i}$ and outputs 1, otherwise, rejects $\sigma_{i}$ and outputs 0.Aggregate-Sign: The Central Hospital produces aggregate signature. The Central Hospital aggregates the verified individual signatures ${{\{\;\sigma }_{i}\;\}}_{i\;=\;1}^{n}$ into a single short signature σ.Aggregate-Verify: The algorithm is responsible for verifying the correctness of the aggregate signature by miner nodes. Inputs aggregate signature σ, all related *DAU_i_*’s trapdoor keys ${{\{\;X}_{i}\;\}}_{i\;=\;1}^{n}$, verifies the correctness of σ. If correct, accepts σ and outputs 1, otherwise, rejects σ and outputs 0.

## 4. The Proposed Multi-Trapdoor Hash Function

The proposed multi-trapdoor hash function based on ECC is presented in this section.

**ParG**: Suppose the security parameter *k*, KGC selects large prime numbers *p*, *q* and elliptic curves over finite fields $y^{2}{\;=\;x}^{3}\;+\;ax\;+\;b\;\mathrm{mod}\;p$, $\;a,\;b\;\in {\;F}_{p}$. Given *G* is a cyclic subgroup of E(*F_p_*), *P* is a *q*-order generator of *G*, KGC takes secure hash function: $\;W\;=\;G\;\to {\;Z}_{q}^{*}$. KGC outputs the system parameter *pars* = (*G*, *P*, *q*, *W*).**KeyG**: Each *DAU_i_* selects randomly trapdoor key $\alpha_{i}\in {\;Z}_{q}^{*}$ and computes hash key: $X_{i\;}{=\;\alpha }_{i\;}P$, then outputs ${{\{\alpha }_{i\;},\;X_{i\;}\}}_{1}^{n}$.**HashG**: Each *DAU_i_* randomly selects the auxiliary parameter *u_i_*, computes trapdoor hash value:
$$T_{i}{\;=\;TH}_{X_{i}}\left(s_{i}{,\;u}_{i}\right)\;=\;W\left({\;s}_{i}{,\;X}_{i}\right)X_{i}{\;+\;u}_{i}P.$$Finally, the Central Hospital calculates multi-trapdoor hash value:
$$T\;=\;\sum_{i=1}^{n}T_{i}$$**TrapColG**: Each *DAU_i_* randomly selects temporary trapdoor key $\beta_{i}\;\in {\;Z}_{q}^{*}$ and computes temporary hash key $Y_{i}\;=\;\beta_{i}P$. The collision parameter is given as
$$u_{i}^{\prime }{\;=\;\alpha }_{i}W\left(s_{i}{,\;X}_{i}\right)\;{-\;\beta }_{i}W\left(s_{i}^{\prime }{,Y}_{i}\right)\;{+\;u}_{i}.$$

Trapdoor collision is one of the properties of trapdoor hash functions [37]. Given hash keys (${\;X}_{i},Y_{i\;}$), trapdoor keys (${\;\alpha }_{i},\;\beta_{i\;}$), message/auxiliary parameter pair $\left(s_{i}{,\;u}_{i}\right)$, and new message $s_{i}^{\prime }$, collision parameter is given by $u_{i}^{\prime }{\;=\;\alpha }_{i}W\left(s_{i}{,\;X}_{i}\right)\;{-\;\beta }_{i}W\left(s_{i}^{\prime }{,Y}_{i}\right)\;{+\;u}_{i}$ which satisfies
$${TH}_{X_{i}}\left(s_{i}{,\;u}_{i}\right){\;=\;TH}_{Y_{i}}\left(s_{i}^{\prime }{,\;u}_{i}^{\prime }\right).$$

That is


$$W\left({\;s}_{i}{,\;X}_{i}\right)X_{i}{\;+\;u}_{i}P\;=\;W\left(s_{i}^{\prime }{,Y}_{i}\right)Y_{i}{\;+\;u}_{i}^{\prime }P$$
$$W\left({\;s}_{i}{,\;X}_{i}\right){\;\alpha }_{i}{\;+\;u}_{i}\;=\;W\left(s_{i}^{\prime }{,Y}_{i}\right){\;\beta }_{i}{\;+\;u}_{i}^{\prime }.$$


From the above proof process, we can conclude that the owner of the trapdoor key can compute the trapdoor collision based on the given input. The proposed multi-trapdoor hash function aggregates multiple trapdoor collisions into one trapdoor collision, which improves the calculation efficiency. On the other hand, people who do not know the trapdoor key cannot calculate the trapdoor collision. Therefore, the proposed multi-trapdoor hash function is secure and efficient to construct the certificateless aggregate signature scheme.

## 5. The Proposed Certificateless Aggregate Signature Scheme

The proposed certificateless aggregate signature scheme based on the multiple trapdoor hash function is presented in this section. We introduce an attribute-based signature [41] and state the information, so that the requirements for medical data in blockchain-based MCPS can be better satisfied.

### 5.1. Setup

In this subsection, KGC will generate the system parameter and send it to data acquisition units *DAU_i_*, patients *P_j_*, and Central Hospitals. Suppose the security parameter *k*, KGC selects large prime numbers *p*, *q* and elliptic curves over finite fields $y^{2}{\;=\;x}^{3}\;+\;ax\;+\;b\;\mathrm{mod}\;p$, $\;a,\;b\;\in {\;F}_{p}$. Given *G* is a cyclic subgroup of E(*F_p_*), *P* is a q-order generator of G, KGC takes seven secure hash functions: ${\;W}_{1}{\;=\;\{\;0,1\;\}}^{*}\to {\;Z}_{q}^{*}$, ${\;W}_{2}\;=\;G\;\to {\;Z}_{q}^{*}$, ${\;W}_{3}{\;=\;\{\;0,1\;\}}^{*}\;\to {\;Z}_{q}^{*}$, $W_{4}\;=\;G\;\to {\;Z}_{q}^{*}$, $W_{5}\;=\;G\;\to {\;Z}_{q}^{*}$, ${\;W}_{6}\;=\;G\;\to {\;Z}_{q}^{*}$, $H\;=\;G\;\to {\;Z}_{q}^{*}$. KGC randomly selects $\lambda \;\in {\;Z}_{q}^{*}$ as the system master key. Then, the public key is $K_{pub}\;=\;\lambda P$. Finally, KGC outputs the system parameter *pars* = (*G*, *P*, *q*, *K_pub_*, *W_1_*, *W_2_*, *W_3_*, *W_4_*, *W_5_*, *W_6_*, *H*).

### 5.2. Pseudonym-Gen

In this phase, the Registry Center calculates the pseudonyms for *DAU_i_* and *P_j_* according to their real identities. The pseudonym system [42] is used to provide conditional privacy protection for doctors, nurses, patients, medical devices, etc. When relevant organizations need to know their real identity, the Registry Center can index their real identity. The Registry Center performs the following procedure to generate pseudonyms for *DAU_i_* and *P_j_*.

The Registry Center accepts *DAU_i_*’s real identity ${RID}_{{DAU}_{i}}$ and calculates its pseudo identity ${ID}_{{DAU}_{i}}^{\prime }{\;=\;W}_{1}{(RID}_{{DAU}_{i}})$. After selecting a random $a_{i}\in {\;Z}_{q}^{*}$, *DAU_i_* calculates $F_{i}{\;=\;a}_{i\;}P$, ${PID}_{{DAU}_{i},1\;}{=\;\lambda W}_{2}{(\;F}_{i}\;)$, and sends ${PID}_{{DAU}_{i},1\;}$ to the Registry Center through the secure channel. The Registry Center calculates ${PID}_{{DAU}_{i},2\;}{=\;W}_{3\;}{(\;ID}_{{DAU}_{i}}^{\prime }{,\;PID}_{{DAU}_{i},1\;})$, and outputs pseudonym ${PID}_{{DAU}_{i}}{\;=\;(PID}_{{DAU}_{i},1\;}{,\;PID}_{{DAU}_{i},2\;})$.The Registry Center accepts *P_j_*’s real identity ${RID}_{P_{j}}$ and calculates its pseudo identity ${ID}_{P_{j}}^{\prime }{\;=\;W}_{1}{(RID}_{P_{j}})$. After selecting a random $b_{j}\in {\;Z}_{q}^{*}$, *P_j_* calculates $E_{j}{\;=\;b}_{j\;}P$, ${PID}_{P_{j},1\;}{=\;\lambda W}_{2}{(E}_{j\;})$, and sends ${PID}_{P_{j},1\;}$ to the Registry Center through the secure channel. The Registry Center calculates ${PID}_{P_{j},2\;}{=\;W}_{3\;}{(\;ID}_{P_{j}}^{\prime }{,\;PID}_{P_{j},1\;})$, and outputs pseudonym ${PID}_{P_{j}}{\;=\;(PID}_{P_{j},1\;}{,\;PID}_{P_{j},2\;})$.

At the same time, the Registry Center builds an index table between the real identities of *DAU_i_* (*P_j_*) and their pseudonyms, such as ${(RID}_{{DAU}_{i\;}}{,\;PID}_{{DAU}_{i}})$, ${(RID}_{P_{j\;}}{,\;PID}_{P_{j}})$, so that when relevant organizations need to know the real identities of *DAU_i_* or *P_j_*, the Registry Center could return their real identities.

### 5.3. DAU_i_ Key-Gen

In this stage, *DAU_i_* completes secret value/public parameter pair generation and sends the public parameter to KGC. With the received public parameter, KGC computes partial private key/partial public key pair. These two key pairs constitute the public keys and private keys of *DAU_i_*. Because the keys of *DAU_i_* are obtained by two entities (KGC and DAU), it is effective to protect the security of the keys.

*DAU_i_* randomly selects the secret value $\alpha_{i}\in {\;Z}_{q}^{*}$, calculates $X_{i\;}{=\;\alpha }_{i\;}P$ as the public parameter. Then, *DAU_i_* sends the public parameter $X_{i\;}$ to the KGC and the Central Hospital.

It then inputs the pseudonym ${PID}_{{DAU}_{i}}$ and public parameters $X_{i\;}$ of *DAU_i_*, KGC randomly selects $\gamma_{i}\in {\;Z}_{q}^{*}$ as the secret value, calculates $V_{i}{\;=\;\gamma }_{i}P$ and *DAU_i_*’s partial private key $\theta_{i}{\;=\;\gamma }_{i}{\;+\;\lambda W}_{4}{(\;PID}_{{DAU}_{i}}{,\;X}_{i}{,\;V}_{i\;})$, then sends $V_{i}$ and $\theta_{i}$ to *DAU_i_* through the secure channel. *DAU_i_* verifies the correctness of partial private key $\theta_{i}$ by checking whether the equation $\theta_{i}{P\;=\;V}_{i}{\;+\;K}_{pub}W_{4}{(PID}_{{DAU}_{i}}{,\;X}_{i}{,\;V}_{i})$ is valid.

*DAU_i_*’s public and private keys are: ${PK}_{{DAU}_{i}}{\;=\;(\;X}_{i}{,\;V}_{i}\;)$, ${\;SK}_{{DAU}_{i}}{\;=\;(\alpha }_{i}{,\;\theta }_{i}\;)$. The partial private key and pseudonym effectively protect *DAU_i_*’s identity information. It plays a role of privacy protection.

### 5.4. Hash-Gen

In this section, each *DAU_i_* generates its own trapdoor hash value and sends it to the Central Hospital. Then, the Central Hospital combines all verified trapdoor hash values into a single value. Based on the trapdoor hash value, the trapdoor collision can be calculated, which can be used to achieve the individual signature.

Firstly, it inputs system parameter *pars*, original message *s_i_*, *DAU_i_*’s hash key (public parameter)${\;X}_{i}$, *DAU_i_* randomly selects auxiliary parameter *u_i_*, and calculates trapdoor hash value $T_{i}{\;=\;TH}_{X_{i}}\left(s_{i}{,\;u}_{i}\right){\;=\;W}_{5}\left({\;s}_{i}{,\;X}_{i}\right)X_{i}{\;+\;u}_{i}P$. Where the original message *s_i_* depends on the attribute value of *DAU_i_*. That is to say, if *DAU_i_* is a doctor or a nurse, then *s_i_* is composed of the ID of the hospital where he or she works, his or her working department, and position titles, etc.; if *DAU_i_* is a medical equipment or app, then *s_i_* is composed of *DAU_i_*’s pseudonym ${PID}_{{DAU}_{i}}$, its manufacturer, categories, the affiliated institutions (hospitals, communities, scientific research institutions, etc.), etc. Using a series of attributes related to the signer to determine their identity can effectively protect the privacy of the signer, such as phone number, home address, email, etc.

When a patient *P_j_* starts data interaction with a *DAU_i_*, the trapdoor hash value *T_i_* of *DAU_i_* is calculated in advance and sent to the Central Hospital. When the treatment of *P_j_* is completed (assuming that *P_j_* generates *n* MRIs with *n DAU_i_*s), the Central Hospital aggregates the trapdoor hash value *T*$\;=\;\sum_{i=1}^{n}T_{i}$ of all the *DAU_i_*s responsible for *P_j_*’s MRIs, and sends *T* to each *DAU_i_*, which interacts with *P_j_*.

### 5.5. Individual-Sign

In this subsection, each *DAU_i_* that provides medical services to the patient *P_j_* completes an individual signature on the medical data for which it is responsible. We define the state information of *DAU_i_* as *Ω_i_*, that is, the pseudonym of *P_j_* associated with this *DAU_i_*. Only the individual signatures with the same *Ω_i_* (that is, for the same patient) can be aggregated.

*DAU_i_* selects the latest timestamp *t_i_* and calculates $\theta_{i}^{\prime }{\;=\;W}_{6}{(\;t}_{i}{,V}_{i}{,\;\Omega }_{i\;})$, $y_{i}{\;=\;\theta }_{i\;}^{\prime }P$. The latest timestamp ensures the timeliness of data collection and resists replay attacks. *DAU_i_* randomly selects temporary trapdoor key $\beta_{i}\in {\;Z}_{q}^{*}$, and calculates the temporary hash key $Y_{i}\;=\;\beta_{i}P$ and the trapdoor hash value ${TH}_{Y_{i}}\left(s_{i}^{\prime }{,\;u}_{i}^{\prime }\right)\;{=\;W}_{5}\left(s_{i}^{\prime }{,Y}_{i}\right)Y_{i}{\;+\;u}_{i}^{\prime }P$. $s_{i}^{\prime }$ represents the digest of *P_j_*’s MRI, which is in the charge of *DAU_i_* during this treatment. According to trapdoor collision (that is ${TH}_{X_{i}}\left(s_{i}{,\;u}_{i}\right){\;=\;TH}_{Y_{i}}\left(s_{i}^{\prime }{,\;u}_{i}^{\prime }\right)$), it calculates collision parameter $u_{i}^{\prime }{\;=\;\alpha }_{i}W_{5}\left(s_{i}{,\;X}_{i}\right)\;{-\;\beta }_{i}W_{5}\left(s_{i}^{\prime }{,Y}_{i}\right)\;{+\;u}_{i}$, $H^{*}{\;=\;H(PID}_{{DAU}_{i\;}}{,\;T\;,\;u}_{i}^{\prime })$, $d_{i}{\;=\;\theta }_{i}^{\prime }{\;-\;(\alpha }_{i}{\;+\;\theta }_{i}{)\;H}^{*}\;\mathrm{mod}\;q$. *DAU_i_*’s individual signature for patient *P_j_* is $\sigma_{i}{\;=\;(y}_{i}{,\;d}_{i})$. *DAU_i_* sends $\left(\sigma_{i}{,\;u}_{i}^{\prime }\right)$ to the Central Hospital.

### 5.6. Individual-Verify

In this stage, the Central Hospital achieves the verification of *DAU_i_*’s individual signature. When the Central Hospital receives *DAU_i_*’s individual signature $\sigma_{i}{\;=\;(y}_{i}{,\;d}_{i})$ and new auxiliary parameter $u_{i}^{\prime }$, the Central Hospital performs the following steps:
Compute $W_{4}^{*}{\;=\;W}_{4}{(PID}_{{DAU}_{i\;}}{,\;X}_{i\;}{,\;V}_{i})$ and ${\;H}^{*}{\;=\;H(PID}_{{DAU}_{i\;}}{,\;T\;,\;u}_{i}^{\prime })$Check whether $d_{i}P\;+\;\left(X_{i}{\;+\;V}_{i}{\;+\;K}_{pub}W_{4}^{*}\right)H^{*}{\;=\;y}_{i}$ holds or not. If it holds, the Central Hospital accepts $\sigma_{i}$ and then stores the encrypted original medical data in the Medical Cloud.

Since $X_{i\;}{=\;\alpha }_{i\;}P$, ${\;\theta }_{i}{P\;=\;V}_{i}{\;+\;K}_{pub}W_{4}{(PID}_{{DAU}_{i}}{,\;X}_{i}{,\;V}_{i})$, $y_{i}{\;=\;\theta }_{i\;}^{\prime }P$, we obtain
$$\begin{array}{l} d_{i}{P\;+(X}_{i}{\;+\;V}_{i}{\;+\;K}_{pub}W_{4}^{*}{)\;H}^{*}\\ {=\;\theta }_{i}^{\prime }P\;-\;\left(\alpha_{i}{\;+\;\theta }_{i}\right)P\;\cdot {\;H}^{*}+\left(X_{i}{\;+\;V}_{i}{\;+\;K}_{pub}W_{4}^{*}\right)H^{*}\\ {=\;\theta }_{i}^{\prime }{P\;-(X}_{i}{\;+\;V}_{i}{\;+\;K}_{pub}W_{4}^{*}{)\;H}^{*}+\left(X_{i}{\;+\;V}_{i}{\;+\;K}_{pub}W_{4}^{*}\right)H^{*}\\ {=\;\theta }_{i}^{\prime }P\\ {=\;y}_{i} \end{array}$$

### 5.7. Aggregate-Sign

In this phase, the Central Hospital aggregates the accepted individual signatures for medical data from the same patient. The Central Hospital checks the status information *Ω_i_* of each *DAU_i_* whose individual signature $\sigma_{i}$ is accepted. For individual signatures with the same *Ω_i_*, the Central Hospital calculates $\omega \;=\;\sum_{i\;=\;1}^{n}y_{i}$, *D*$\;=\;\sum_{i\;=\;1}^{n}d_{i}$, and the aggregate signature σ = (ω, D). Then, the Central Hospital forms a transaction by *P_j_*’s MRI digest, aggregation signature, access control, and the specific location of the original medical data in the Medical Cloud. Finally, a transaction request is sent to the Medical Chain.

### 5.8. Aggregate-Verify

After the miner receives the message, the aggregate signature is verified through the consensus mechanism. If the equation *DP*$\;+\sum_{i=1}^{n}\left(X_{i}{\;+\;V}_{i}{\;+\;K}_{pub}W_{4}^{*}\right){\;H}^{*}=\;\omega$ holds, the information is broadcast to other nodes in the network. The other nodes start consensus verification of the transaction and broadcast on the network. After the verification is successful, the transaction is added to the block.

## 6. Security Discussion

### 6.1. Correctness Proof

The correctness proof of the aggregate is verified as follows:
$$\begin{array}{l} \;\;\;DP+\sum_{i=1}^{n}\left(X_{i}{\;+\;V}_{i}{\;+\;K}_{pub}W_{4}^{*}\right){\;H}^{*}\\ =\sum_{i=1\;}^{n}\left(\theta_{i}^{\prime }P\;-\;\left(\alpha_{i}{\;+\;\theta }_{i}\right)P\;\cdot {\;H}^{*}\right)+\sum_{i=1}^{n}\left(X_{i}{\;+\;V}_{i}{\;+\;K}_{pub}W_{4}^{*}\right){\;H}^{*}\\ =\sum_{i=1}^{n}{[\theta }_{i}^{\prime }P\;-\;\left(X_{i}{\;+\;V}_{i}{\;+\;K}_{pub}W_{4}^{*}\right){\;H}^{*}]\;+\sum_{i=1}^{n}\left(X_{i}{\;+\;V}_{i}{\;+\;K}_{pub}W_{4}^{*}\right){\;H}^{*}\\ =\;\sum_{i=1}^{n}\theta_{i\;}^{\prime }P\\ =\;\sum_{i=1}^{n}y_{i}\\ =\;\omega \end{array}$$

### 6.2. Security Proof

**Theorem** **1.**
*In the random oracle model, the proposed certificateless aggregate signature scheme is existentially unforgeable against adaptive chosen-message attacks under the assumption that the ECDLP problem is hard.*


This theorem is obtained by combining Lemmas 1 and 2.

**Lemma** **1.***Given an***A****_Ⅰ_***type adversary*$C_{1}$*makes at most*$q_{S}$*Sign queries,*$q_{K}$*Partial-Key-Gen queries,*$q_{SK}$*Partial-Key-Gen queries within a period t in the random oracle model, and wins the game with an non-negligible probability ε, that is, successfully forging the signature of the proposed scheme. Then, an algorithm*$T_{1}$*can be performed in polynomial time, and solve an instance of ECDLP with probability (supposing the number of aggregate signatures is n)*$\varepsilon^{\prime }\;\geq \;\frac{\varepsilon }{{ne\;(q}_{S}\;+\;n)}\;(1\;-\;\frac{q_{K}}{2^{t}})\;(1\;-\;\frac{q_{SK}}{2^{t}})$.

**Proof.** Suppose $T_{1}$ is a solution of ECDLP and (P, xP) ϵ G as an instance of ECDLP, the goal of the algorithm $T_{1}$ is to compute *x*. Suppose $T_{1}$ makes $q_{S}$ Sign queries on $q_{S}$ identities, and generates n aggregate signatures at the challenge stage, $T_{1}$ selects ${PID}_{{DAU}_{k}\;}$ as the target victim, and the probability of the selection is $\mu \in \;[\frac{1}{q_{S}\;+\;n},\;\frac{1}{q_{S}\;+\;1}]$. We set up a game between adversary $C_{1}$ and challenger $Z_{1}$, and the detailed interaction process is as follows:**Setup**: Given $K_{pub\;}=\;xP$, challenger $Z_{1}$ inputs security parameters *k*, generates system parameter *pars* = (*G*, *P*, *q*, *K_pub_*, *W_1_*, *W_2_*, *W_3_*, *W_4_*, *W_5_*, *W_6_*, *H*), and sends *pars* to adversary $C_{1}$. $Z_{1}$ needs to maintain nine lists ${(\;L}_{W_{4}}{,\;L}_{W_{5}}{,\;L}_{W_{6}}{,\;L}_{H}{,\;L}_{P}{,\;L}_{PK}{,L}_{SK}{,\;L}_{T}{,\;L}_{S})$, whose initial values are empty.**Query**: $C_{1}$ adaptively performs the following oracle queries.
*W_4_ hash query*: When $C_{1}$ makes a *W_4_* hash query with parameter ${(PID}_{{DAU}_{i}\;}{,\;X}_{i}{,\;V}_{i})$, $Z_{1}$ checks whether existing ${(PID}_{{DAU}_{i}\;}{,\;X}_{\;i}{,\;V}_{i}{,\;\delta }_{W_{4}})\;\in {\;L}_{W_{4}}$ or not, if so, $Z_{1}$ sends ${\;\delta }_{W_{4}}$ to $C_{1}$. Otherwise, $Z_{1}$ selects a random ${\;\delta }_{W_{4}}\in Z_{q}^{*}$. If the list ${\;L}_{W_{4}}$ does not include the tuple (*, *, *, ${\;\delta }_{W_{4}}$), $Z_{1}$ sends ${\;\delta }_{W_{4}}$ to $C_{1}$ and saves ${(PID}_{{DAU}_{i}\;}{,\;X}_{i}{,\;V}_{i}{,\;\delta }_{W_{4}})\;$ into the hash list ${\;L}_{W_{4}}$.*W_5_ hash query*: When $C_{1}$ makes a *W_5_* hash query with parameter ${(s}_{i\;}{,\;X}_{i})$, $Z_{1}$ checks whether existing ${(s}_{i\;}{,\;X}_{i\;}{,\;\delta }_{W_{5}})\in {\;L}_{W_{5}}$ or not, if so, $Z_{1}$ sends ${\;\delta }_{W_{5}}$ to $C_{1}$. Otherwise, $Z_{1}$ selects a random ${\;\delta }_{W_{5}}\in Z_{q}^{*}$. If the list ${\;L}_{W_{5}}$ does not include the tuple (*, *, ${\;\delta }_{W_{5}}$), $Z_{1}$ sends ${\;\delta }_{W_{5}}$ to $C_{1}$ and saves ${(s}_{i\;}{,\;X}_{i\;}{,\;\delta }_{W_{5}})$ into the hash list ${\;L}_{W_{5}}$.*W_6_ hash query*: When $C_{1}$ makes a *W_6_* hash query with parameter ${(t}_{i\;}{,\;V}_{i\;}{,\;\Omega }_{i})$, $Z_{1}$ checks whether existing ${(t}_{i\;}{,\;V}_{i\;}{,\;\Omega }_{i}{,\;\delta }_{W_{6}})\in {\;L}_{W_{6}}$ or not, if so, $Z_{1}$ sends ${\;\delta }_{W_{6}}$ to $C_{1}$. Otherwise, $Z_{1}$ selects a random ${\;\delta }_{W_{6}}\in Z_{q}^{*}$. If the list ${\;L}_{W_{6}}$ does not include the tuple (*, *, *, ${\;\delta }_{W_{6}}$), $Z_{1}$ sends $\delta_{W_{6}}$ to $C_{1}$ and saves ${(t}_{i\;}{,\;V}_{i\;}{,\;\Omega }_{i}{,\;\delta }_{W_{6}})$ into the hash list ${\;L}_{W_{6}}$.*H hash query*: When $C_{1}$ makes an *H* hash query with parameter ${(PID}_{{DAU}_{i\;}}{,T,\;u}_{i}^{\prime })$, $Z_{1}$ checks whether existing ${(PID}_{{DAU}_{i\;}}{,T,\;u}_{i}^{\prime }{,\;\delta }_{H})\in {\;L}_{H}$ or not, if so, $Z_{1}$ sends ${\;\delta }_{H}$ to $C_{1}$. Otherwise, then $Z_{1}$ selects a random ${\;\delta }_{H}\in Z_{q}^{*}$. If the list $L_{H}$ does not include the tuple (*, *, *, ${\;\delta }_{H}$), $Z_{1}$ sends ${\;\delta }_{H}$ to $C_{1}$ and saves ${(PID}_{{DAU}_{i\;}}{,\;T,\;u}_{i}^{\prime }{,\;\delta }_{H})$ into the hash list ${\;L}_{H}$.*Partial-Key-Gen query*: When $C_{1}$ makes a Partial-Key-Gen query with parameter ${(PID}_{{DAU}_{i\;}}{,\;X}_{i\;})$, $Z_{1}$ checks whether existing ${(\;PID}_{{DAU}_{i}\;}{,\;\theta }_{i\;}{,\;V}_{i}\;)\in {\;L}_{P}$ or not.-If *L_P_* includes the tuple ${(\;PID}_{{DAU}_{i}\;}{,\;\theta }_{i\;}{,\;V}_{i}\;)$, $Z_{1}$ sends ${(\;\theta }_{i\;},V_{i})$ to $C_{1}$.-If *L_P_* does not include the tuple ${(\;PID}_{{DAU}_{i}\;}{,\;\theta }_{i\;}{,\;V}_{i}\;)$ and ${PID}_{{DAU}_{i}\;}\neq {\;PID}_{{DAU}_{k}}$, $Z_{1}$ selects a random $\theta_{i\;}{,\;\delta }_{W_{4}}\in Z_{q}^{*}$, computes $V_{i}{\;=\;\theta }_{i\;}{P\;-\;K}_{pub}{\;\delta }_{W_{4}}$, sends ${(\;\theta }_{i\;},V_{i})$ to $C_{1}$ and saves ${(\;PID}_{{DAU}_{i}\;}{,\;\theta }_{i\;}{,\;V}_{i}\;)$ into the hash list $L_{P}$. If list $L_{W_{4}}$ does not include corresponding tuple, then $Z_{1}$ adds tuple ${(PID}_{{DAU}_{i\;}}{,\;X}_{i\;}{,\;V}_{i\;}{,\;\delta }_{W_{4}})$ into $L_{W_{4}}$.-If *L_P_* does not include the tuple ${(\;PID}_{{DAU}_{i}\;}{,\;\theta }_{i\;}{,\;V}_{i}\;)$ and ${PID}_{{DAU}_{i}\;}{=\;PID}_{{DAU}_{k}}$, $Z_{1}$ randomly selects $\theta_{i}{,\;\delta }_{W_{4}}\in Z_{q}^{*}$, lets $V_{k}{\;=\;\gamma }_{r}P$ ($\gamma_{r}\in Z_{q}^{*}$ is a known random number to $Z_{1}$ ), then saves ${(\;PID}_{{DAU}_{k\;}}{,\;\theta }_{k\;}{,\;V}_{k}\;)$ into the hash list $L_{P}$ and sends ${(\;\theta }_{k\;}{,\;V}_{k})$ to $C_{1}.$ If list $L_{W_{4}}$ does not include corresponding tuple, then $Z_{1}$ adds tuple ${(PID}_{{DAU}_{k\;}}{,\;X}_{k\;}{,\;V}_{k\;}{,\;\delta }_{W_{4}})$ into $L_{W_{4}}$.*Secret-Key-Gen query*: Suppose that the query is on a pseudo identity ${PID}_{{DAU}_{i}\;}$.If the list $L_{SK}$ includes ${\;(PID}_{{DAU}_{i\;}}{,\;\alpha }_{i\;}{,\;\theta }_{i})$, $Z_{1}$ sends ${\;(\alpha }_{i\;}{,\;\theta }_{i\;})$ to $C_{1}.$ Otherwise, $Z_{1}$ selects a random $\alpha_{i}\in Z_{q}^{*}$ and computes $X_{i}{\;=\;\alpha }_{i\;}P$. Then $Z_{1}$ makes a Partial-Key-Gen query by ${(PID}_{{DAU}_{i\;}}{,\;X}_{i})$ and adds ${(PID}_{{DAU}_{i}\;}{,\;\alpha }_{i\;}{,\;\theta }_{i})$ into list $L_{SK}$. $Z_{1}$ sends ${(\alpha }_{i\;}{,\;\theta }_{i\;})$ to $C_{1}$ and adds ${(PID}_{{DAU}_{i\;}}{,\;X}_{i\;}{,\;V}_{i})$ into list $L_{PK}$.*Public-Key-Gen query*: Suppose that the query is on a pseudo identity ${PID}_{{DAU}_{i}\;}$.If the list $L_{PK}$ includes ${\;(PID}_{{DAU}_{i}\;}{,\;X}_{i\;}{,\;V}_{i}\;)$, $Z_{1}$ sends ${\;(\;X}_{i\;}{,\;V}_{i}\;)$ to $C_{1}.$ Otherwise, $Z_{1}$ selects a random $\alpha_{i}\in Z_{q}^{*}$ and computes $X_{i}{\;=\;\alpha }_{i}P$. Then $Z_{1}$ makes a *Partial- Key query* by ${(PID}_{{DAU}_{i}\;}{,\;X}_{i})$ and adds ${(PID}_{{DAU}_{i}\;}{,\;X}_{i\;}{,\;V}_{i}\;)$ into list $L_{PK}$. $Z_{1}$ sends ${(\;X}_{i\;}{,\;V}_{i}\;)$ to $C_{1}$ and adds ${(PID}_{{DAU}_{i\;}}{,\;\alpha }_{i\;}{,\;\theta }_{i})$ into list $L_{SK}$.*Public-Key-Replacement query*: $C_{1}$ can select a new public key ${PK}_{{DAU}_{i}}^{*}{\;=\;(\;X}_{i}^{*}{,\;V}_{i}^{*}\;)$ to replace the original public key ${PK}_{{DAU}_{i}}$ of any legitimate *DAU_i_*.*Hash-Gen**query*: When $C_{1}$ makes a Hash-Gen query with parameter ${(s}_{i\;}{,\;u}_{i\;})$, $Z_{1}$ checks whether existing ${(s}_{i\;}{,\;u}_{i\;}{,\;T}_{i})\in {\;L}_{T}\;$ or not, if so, $Z_{1}$ returns *T_i_* to $C_{1}$. Otherwise, selects a random $\alpha_{i}\;\in {\;Z}_{q}^{*}$ and computes:
$$T_{i}{\;=\;W}_{5}\left(s_{i\;}{,\;\alpha }_{i\;}P\right){\;\alpha }_{i\;}{P\;+\;u}_{i\;}P.$$Sends *T_i_* to $C_{1}$ and saves ${(s}_{i\;}{,\;u}_{i\;}{,\;T}_{i})$ into the hash list ${\;L}_{T}$.*Sign query*: When $C_{1}$ makes a sign query with parameter ${(\alpha }_{i\;}{,\;\Omega }_{i\;}{,\;s}_{i\;}{,\;s}_{i}^{\prime })$, $Z_{1}$ checks whether ${PID}_{{DAU}_{i}}{\;=\;PID}_{{DAU}_{k}}$ or not, if so, $Z_{1}$ randomly selects $t_{i}\in {\;Z}_{q}^{*}$ and $\beta_{i}\in {\;Z}_{q}^{*}$, and computes:
$$\begin{array}{l} \theta_{i}^{\prime }{\;=\;W}_{6}{(t}_{i\;}{,\;V}_{i\;}{,\;\Omega }_{i})\\ y_{i}{\;=\;\theta }_{i\;}^{\prime }P\\ Y_{i}\;=\;\beta_{i}P\\ H^{*}\;=\;H\left({PID}_{{DAU}_{i\;}}{,\;T\;,\;u}_{i}^{\prime }\right)\\ u_{i}^{\prime }{\;=\;\alpha }_{i}W_{5}\left(s_{i\;}{,\;X}_{i}\right)\;{-\;\beta }_{i}W_{5}\left(s_{i\;}^{\prime }{,Y}_{i}\right){\;+\;u}_{i}\\ d_{i}{\;=\;\theta }_{i}^{\prime }{\;-\;(\alpha }_{i}{\;+\;\theta }_{i}{)\;H}^{*}\;\mathrm{mod}\;q \end{array}$$Then, $Z_{1}$ generates individual signature ${(y}_{i}{,\;d}_{i})$ and sends it to $C_{1}$.Otherwise, $Z_{1}$ outputs failure and halts.*Aggregate-Sign query*: When all of the ${PID}_{{DAU}_{i}}\;(1\;\leq \;i\;\leq \;n)$ satisfies ${PID}_{{DAU}_{i}}\;\neq {\;PID}_{{DAU}_{k}}$, $Z_{1}$ randomly selects $t_{i}\in {\;Z}_{q}^{*}$ and $\beta_{i}\in {\;Z}_{q}^{*}$ for every *DAU_i_*
(1 ≤ i ≤ n). Then $Z_{1}$ calculates
$$\begin{array}{l} \theta_{i}^{\prime }{\;=\;W}_{6}{(t}_{i\;}{,\;V}_{i\;}{,\;\Omega }_{i})\\ y_{i}{\;=\;\theta }_{i\;}^{\prime }P\\ Y_{i}\;=\;\beta_{i}P\\ H^{*}\;=\;H\left({PID}_{{DAU}_{i\;}}{,\;T\;,\;u}_{i}^{\prime }\right)\\ u_{i}^{\prime }{\;=\;\alpha }_{i}W_{5}\left(s_{i\;}{,\;X}_{i}\right)\;{-\;\beta }_{i}W_{5}\left(s_{i\;}^{\prime }{,Y}_{i}\right){\;+\;u}_{i}\\ d_{i}{\;=\;\theta }_{i}^{\prime }{\;-\;(\alpha }_{i}{\;+\;\theta }_{i}{)\;H}^{*}\;\mathrm{mod}\;q\\ \omega \;=\;\sum_{i=1}^{n}y_{i}\\ D=\;\sum_{i=1}^{n}d_{i} \end{array}$$Then, $Z_{1}$ generates aggregate signature (*ω*, *D*) and sends it to $C_{1}$.Otherwise, if ${PID}_{{DAU}_{i}}{\;=\;PID}_{{DAU}_{k}}$, $Z_{1}$ outputs failure and halts.*Individual-Verify query*: When $C_{1}$ makes an Individual-Verify query, $Z_{1}$ checks whether the corresponding tuple of ${PID}_{{DAU}_{i}}$ is included in list *L_PK_*.
-If the corresponding tuple of ${PID}_{{DAU}_{i}}$ is included in list *L_PK_* and ${PID}_{{DAU}_{i}}\;\neq {\;PID}_{{DAU}_{k}}$, $Z_{1}$ calculates $W_{4}^{*}{\;=\;W}_{4}{(PID}_{{DAU}_{i\;}}{,\;X}_{i\;}{,\;V}_{i})$, $H^{*}\;=\;H\left({PID}_{{DAU}_{i\;}}{,\;T\;,\;u}_{i}^{\prime }\right)$ and verifies whether the equation $d_{i}{P\;=\;y}_{i}+\;\left(X_{i}{\;+\;V}_{i}{\;+\;K}_{pub}W_{4}^{*}\right)H^{*}$ holds or not, if so, $Z_{1}$ returns 1 to $C_{1}$, otherwise, returns 0 to $C_{1}$.-If the corresponding tuple of ${PID}_{{DAU}_{i}}$ is included in list *L_PK_* and ${PID}_{{DAU}_{i}}{\;=\;PID}_{{DAU}_{k}}$, $Z_{1}$ returns 1 to $C_{1}$ when the list *L_H_* includes the tuple $\left({PID}_{{DAU}_{i\;}}{,\;T,\;u}_{i}^{\prime }{,\;\delta }_{H}\right)$, otherwise, $Z_{1}$ returns 0 to $C_{1}.$-If the corresponding tuple of ${PID}_{{DAU}_{i}}$ is not included in list *L_PK_*, $Z_{1}$ returns 1 to $C_{1}$ when the list *L_H_* includes the tuple $\left({PID}_{{DAU}_{i\;}}{,\;T,\;u}_{i}^{\prime }{,\;\delta }_{H}\right)$, otherwise, $Z_{1}$ returns 0 to $C_{1}.$**Forge**: After the above polynomial bounded queries, $Z_{1}$ outputs the aggregate signature *σ^*^* = (*ω^*^*, *D^*^*) of ${PID}_{{DAU}_{i}}\;(1\;\leq \;i\;\leq \;n)$, in which at least one ${PID}_{{DAU}_{i\;}}(i\;\in \;[1,\;n])$ does not make Partial-Key-Gen query and Secret-Key-Gen query, and at least one message $s_{i\;}^{\prime }(i\;\in \;[1,\;n])$ does not make Sign query.If all the ${PID}_{{DAU}_{i}}\;(1\;\leq \;i\;\leq \;n)$ satisfies ${PID}_{{DAU}_{i}}\;\neq {\;PID}_{{DAU}_{k}}$, then $Z_{1}$ outputs failure and halts. Otherwise, if one ${PID}_{{DAU}_{i}}\;(1\;\leq \;i\;\leq \;n)$ satisfies ${PID}_{{DAU}_{i}}{\;=\;PID}_{{DAU}_{k}}$, then $Z_{1}$ queries the corresponding tuples of ${PID}_{{DAU}_{i}}\;(1\;\leq \;i\;\leq \;n)$ in the lists *L_PK_*, *L_SK_*, *L_H_* and checks whether the equation *DP*$+\sum_{i=1}^{n}\left(X_{i}{\;+\;V}_{i}{\;+\;K}_{pub}W_{4}^{*}\right){\;H}^{*}=\;\omega \;$ holds or not:
-If the equation holds, $Z_{1}$ outputs *x*${=\;(W}_{4}^{*}H^{*}{)}^{-1}\{\;\sum_{\;i=1,i\neq k\;}^{\;n}\left[{\;\theta }_{i\;}^{\prime }-\;\left(\alpha_{i}{\;+\;\theta }_{i}\right)H^{*}\;\right]{\;+\;\theta }_{k\;}^{\prime }-\;\left(\alpha_{k}{\;+\;\gamma }_{r\;}\right)H^{*\;}-\;D\;\}$ as the efficient solution to the ECDLP.-Otherwise, $Z_{1}$ cannot solve the discrete logarithmic problem, because:
$$\begin{array}{ll} D & =\;\sum_{i=1}^{n}d_{i}\\ & =\sum_{i=1\;}^{n}\left[{\;\theta }_{i}^{\prime }\;-\;\left(\alpha_{i}{\;+\;\theta }_{i}\right)\;\cdot {\;H}^{*\;}\right]\\ & =\sum_{i=1,i\neq k\;}^{n}\left[{\;\theta }_{i}^{\prime }\;-\;\left(\alpha_{i}{\;+\;\theta }_{i}\right)\;\cdot {\;H}^{*\;}\right]{+\theta }_{k}^{\prime }\;-\;\left(\alpha_{k}{\;+\;\theta }_{k}\right)\;\cdot {\;H}^{*}\\ & =\sum_{i=1,i\neq k\;}^{n}\left[{\;\theta }_{i}^{\prime }\;-\;\left(\alpha_{i}{\;+\;\theta }_{i}\right)\;\cdot {\;H}^{*\;}\right]{+\theta }_{k}^{\prime }\;-\;\left(\alpha_{k}{\;+\;\gamma }_{r\;}{+\;\mathrm{xW}}_{4}^{*}\right)\;\cdot {\;H}^{*} \end{array}$$If $C_{1}$ queries all ${PID}_{{DAU}_{i}}\;(1\;\leq \;i\;\leq \;n)$ with Partial-Key-Gen and Secret-Key-Gen, $Z_{1}$ will terminate the simulation. Suppose that
Event *E_1_* represents that at least a ${PID}_{{DAU}_{k}}\;(1\;\leq \;k\;\leq \;n)$ does not make Partial-Key-Gen query and Secret-Key-Gen query.Event *E_2_* represents that $Z_{1}$ does not terminate at the Sign-query stage.Event *E_3_* represents that $Z_{1}$ does not terminate at the challenge stage.The probability of solving the ECDLP by algorithm $T_{1}$ is as follows:
$$\begin{array}{l} \mathrm{Pr}\left[{\;E}_{1\;}\right]\;\geq \;\frac{1}{n}\;(1\;-\;\frac{q_{K}}{2^{t}})(1\;-\;\frac{q_{SK}}{2^{t}})\\ \mathrm{Pr}\left[{\;E}_{2\;}{|\;E}_{1}\right]{\;=\;(1\;-\;\varphi )}^{q_{S}}\\ {\mathrm{Pr}[\;E}_{2\;}\wedge {\;E}_{1\;}]=\mathrm{Pr}\left[{\;E}_{2\;}{|\;E}_{1}\right]\mathrm{Pr}\left[{\;E}_{1\;}\right]\geq \;\frac{1}{n}\;(1\;-\;\frac{q_{K}}{2^{t}})(1\;-\;\frac{q_{SK}}{2^{t}}{)(1\;-\;\varphi )}^{q_{S}}\\ \mathrm{Pr}\left[{\;E}_{3\;}\right]\;=\;\mu \end{array}$$The probability that $Z_{1}$ does not terminate during the whole simulation is at least$$\frac{1}{n}\;(1\;-\;\frac{q_{K}}{2^{t}})(1\;-\;\frac{q_{SK}}{2^{t}}{)(1\;-\;\varphi )}^{q_{S}}\;\mu$$Since $\mu \in \;[\frac{1}{q_{S}\;+\;n},\;\frac{1}{q_{S}\;+\;1}]$, when $q_{S}$ is large enough, ${\left(1\;-\;\varphi \right)}^{q_{S}}$ tends to ${e\;}^{-1}$, so the probability that $Z_{1}$ does not terminate during the simulation is at least
$$\frac{1}{{ne\;(q}_{S}\;+\;n)}\;(1\;-\;\frac{q_{K}}{2^{t}})(1\;-\;\frac{q_{SK}}{2^{t}})$$
In summary, if $Z_{1}$ is not terminated during the simulation, and $C_{1}$ breaks the unforgeability of the proposed scheme with a non-negligible probability ε, $T_{1}$ can successfully solve ECDLP with a non-negligible probability:
$$\varepsilon^{\prime }\geq \frac{\varepsilon }{{ne\;(q}_{S}\;+\;n)}\;(1\;-\;\frac{q_{K}}{2^{t}})\;(1\;-\;\frac{q_{SK}}{2^{t}})$$ □

**Lemma** **2.***Given an***A****_Ⅱ_***type adversary*$C_{2}$*makes at most*$q_{S}$*Sign queries,*$q_{K}$*Partial-Key-Gen queries,*$q_{SK}$*Partial-Key-Gen queries within a period t in the random oracle model, and wins the game with an non-negligible probability ε, that is, successfully forging the signature of the proposed scheme. Then, an algorithm*$T_{2}$*can be performed in polynomial time, and solve an instance of ECDLP with probability (supposing the number of aggregate signatures is n)*$\varepsilon^{\prime }\;\geq \;\frac{\varepsilon }{{ne\;(q}_{S}\;+\;n)}\;(1\;-\;\frac{q_{K}}{2^{t}})\;(1\;-\;\frac{q_{SK}}{2^{t}})$.

**Proof.** Suppose $T_{2}$ is a solution of ECDLP and (P, xP) ϵ G as an instance of ECDLP. The goal of the algorithm $T_{2}$ is to compute *x*. $T_{2}$ selects ${PID}_{{DAU}_{k}}$ as the target victim, and the probability of the selection is $\mu \in \;[\frac{1}{q_{S}\;+\;n},\;\frac{1}{q_{S}\;+\;1}]$. We set up a game between adversary $C_{2}$ and challenger $Z_{2}$, and the detailed interaction process is as follows:**Setup**: Challenger $Z_{2}$ inputs security parameters *k*, generates system parameter *pars*, and sends *pars* = (*G*, *P*, *q*, *K_pub_*, *W_1_*, *W_2_*, *W_3_*, *W_4_*, *W_5_*, *W_6_*, *H*) to adversary $C_{2}$. $Z_{2}$ needs to maintain nine lists ${(\;L}_{W_{4}}{,\;L}_{W_{5}}{,\;L}_{W_{6}}{,\;L}_{H}{,\;L}_{P}{,\;L}_{PK}{,L}_{SK,}{\;L}_{T}{,\;L}_{S})$, whose initial values are empty.**Query**: Adversary $C_{2}$ makes the same queries as that of *W_4_* hash, *W_5_* hash, *W_6_* hash, *H* hash, Secret-Key-Gen, Public-Key-Gen, Hash-Gen, Sign query, Aggregate-Sign query in Lemma 1.
*Partial-Key-Gen query*: When $C_{2}$ makes a Partial-Key-Gen query with parameter ${(PID}_{{DAU}_{i\;}}{,\;X}_{i\;})$, $Z_{2}$ checks whether existing ${(\;PID}_{{DAU}_{i}\;}{,\;\theta }_{i\;}{,\;V}_{i}\;)\in {\;L}_{P}$ or not.
-If the tuple ${(\;PID}_{{DAU}_{i}\;}{,\;\theta }_{i\;}{,\;V}_{i}\;)$ is included in the list *L_P_*, $Z_{2}$ sends ${(\;\theta }_{i\;},V_{i})$ to $C_{2}$.-If the tuple ${(\;PID}_{{DAU}_{i}\;}{,\;\theta }_{i\;}{,\;V}_{i}\;)$ is not included in the list *L_P_* and ${PID}_{{DAU}_{i}\;}\neq {\;PID}_{{DAU}_{k}}$, $Z_{2}$ selects a random $\theta_{i\;}{,\;\delta }_{W_{4}}\in Z_{q}^{*}$, computes $V_{i}{\;=\;\theta }_{i\;}{P\;-\;K}_{pub}{\;\delta }_{W_{4}}$, sends ${(\;\theta }_{i\;},V_{i})$ to $C_{2}$ and saves ${(\;PID}_{{DAU}_{i}\;}{,\;\theta }_{i\;}{,\;V}_{i}\;)$ into the list $L_{P}$. Then $Z_{1}$ adds tuple ${(PID}_{{DAU}_{i\;}}{,\;X}_{i\;}{,\;V}_{i\;}{,\;\delta }_{W_{4}})$ into $L_{W_{4}}$.-If the tuple ${(\;PID}_{{DAU}_{i}\;}{,\;\theta }_{i\;}{,\;V}_{i}\;)$ is not included in the list *L_P_* and ${PID}_{{DAU}_{i}\;}{=\;PID}_{{DAU}_{k}}$, $Z_{2}$ randomly selects $\theta_{i}{,\;\delta }_{W_{4}}\in Z_{q}^{*}$, lets $V_{k}\;=\;xP$, then saves ${(PID}_{{DAU}_{k\;}}{,\;\theta }_{k\;}{,\;V}_{k}\;)$ into the hash list $L_{P}$ and sends ${(\;\theta }_{k\;}{,\;V}_{k})$ to $C_{2}.$ Then $Z_{2}$ adds tuple ${(PID}_{{DAU}_{k\;}}{,\;X}_{k\;}{,\;V}_{k\;}{,\;\delta }_{W_{4}})$ into $L_{W_{4}}$.*Individual-Verify query*: When $C_{2}$ makes an Individual-Verify query with parameter ${(PID}_{{DAU}_{i\;}}{,\;s}_{i\;}^{\prime })$, $Z_{2}$ checks whether the corresponding tuple of ${PID}_{{DAU}_{i}}$ is included in list *L_PK_*.
-If the corresponding tuple of ${PID}_{{DAU}_{i}}$ is included in list *L_PK_* and ${PID}_{{DAU}_{i}}\;\neq {\;PID}_{{DAU}_{k}}$, $Z_{2}$ calculates $W_{4}^{*}{\;=\;W}_{4}{(PID}_{{DAU}_{i\;}}{,\;X}_{i\;}{,\;V}_{i})$, $H^{*}\;=\;H\left({PID}_{{DAU}_{i\;}}{,\;T\;,\;u}_{i}^{\prime }\right)$ and verifies whether the equation $d_{i}P+\;\left(X_{i}{\;+\;V}_{i}{\;+\;K}_{pub}W_{4}^{*}\right)H^{*}{\;=\;y}_{i}\;$ holds or not, if so, $Z_{2}$ returns 1 to $C_{2}$, otherwise, returns 0 to $C_{2}$.-If the corresponding tuple of ${PID}_{{DAU}_{i}}$ is included in list *L_PK_* and ${PID}_{{DAU}_{i}}{\;=\;PID}_{{DAU}_{k}}$, $Z_{2}$ returns 1 to $C_{2}$ when the list *L_H_* includes the tuple $\left({PID}_{{DAU}_{i\;}}{,\;T,\;u}_{i}^{\prime }{,\;\delta }_{H}\right)$, otherwise, $Z_{2}$ returns 0 to $C_{2}.$**Forge**: After the above polynomial bounded queries, $Z_{2}$ outputs the aggregate signature *σ^*^* = (*ω^*^*, *D^*^*) of ${PID}_{{DAU}_{i}}\;(1\;\leq \;i\;\leq \;n)$, in which at least one ${PID}_{{DAU}_{i\;}}(i\;\in \;[1,\;n])$ does not perform the Partial-Key-Gen query and Secret-Key-Gen query, and at least one message, $s_{i\;}^{\prime }(i\;\in \;[1,\;n])$ does not make Sign query.If all the ${PID}_{{DAU}_{i}}\;(1\;\leq \;i\;\leq \;n)$ satisfy ${PID}_{{DAU}_{i}}\;\neq {\;PID}_{{DAU}_{k}}$, then $Z_{2}$ outputs failure and halts. Otherwise, if one ${PID}_{{DAU}_{K}}\;(1\;\leq \;K\;\leq \;n)$ satisfies ${PID}_{{DAU}_{K}}{\;=\;PID}_{{DAU}_{k}}$, then $Z_{2}$ queries the corresponding tuples of ${PID}_{{DAU}_{i}}\;(1\;\leq \;i\;\leq \;n)$ in the lists *L_PK_*, *L_SK_*, *L_H_*, $L_{W_{4}}$ and checks whether the equation *DP*$\;+\sum_{i=1}^{n}\left(X_{i}{\;+\;V}_{i}{\;+\;K}_{pub}W_{4}^{*}\right){\;H}^{*}=\;\omega$ holds or not:
-If the equation holds, $Z_{2}$ outputs *x*${\;=\;(H}^{*}{)}^{-1}\{\;\sum_{\;i=1,i\neq k\;}^{\;n}\left[{\;\theta }_{i\;}^{\prime }-\;\left(\alpha_{i}{\;+\;\theta }_{i}\right)H^{*}\right]{\;+\;\theta }_{k\;}^{\prime }-\;\left(\alpha_{k}{\;+\;\lambda W}_{4}^{*}\right)H^{*\;}-\;D\;\}$ as the solution to the ECDLP.-Otherwise, $Z_{2}$ cannot solve the discrete logarithmic problem, because:
$$\begin{array}{ll} D & =\;\sum_{i=1}^{n}d_{i}\\ & =\sum_{i=1\;}^{n}\left[\theta_{i}^{\prime }\;-\;\left(\alpha_{i}{\;+\;\theta }_{i}\right)\;\cdot {\;H}^{*}\right]\\ & =\sum_{i=1,i\neq k\;}^{n}\left[\theta_{i}^{\prime }\;-\;\left(\alpha_{i}{\;+\;\theta }_{i}\right)\;\cdot {\;H}^{*}\right]{+\theta }_{k}^{\prime }\;-\;\left(\alpha_{k}{\;+\;\theta }_{k}\right)\;\cdot {\;H}^{*}\\ & =\sum_{i=1,\;i\neq k\;}^{n}\left[\theta_{i}^{\prime }\;-\;\left(\alpha_{i}{\;+\;\theta }_{i}\right)\;\cdot {\;H}^{*}\right]{+\theta }_{k}^{\prime }\;-\;\left(\alpha_{k}{\;+\;x\;+\;\lambda W}_{4}^{*}\right)\;\cdot {\;H}^{*} \end{array}$$It can be seen from the proof of Lemma 1 that the probability that $Z_{2}$ does not terminate during the simulation is at least$$\frac{1}{{ne\;(q}_{S}\;+\;n)}\;(1\;-\;\frac{q_{K}}{2^{t}})(1\;-\;\frac{q_{SK}}{2^{t}})$$Therefore, if $Z_{2}$ is not terminated during the simulation, and $C_{2}$ breaks the unforgeability of the proposed scheme with a non-negligible probability, $T_{2}$ can successfully solve ECDLP with a non-negligible probability:
$$\varepsilon^{\prime }\geq \frac{\varepsilon }{{ne\;(q}_{S}\;+\;n)}\;(1\;-\;\frac{q_{K}}{2^{t}})\;(1\;-\;\frac{q_{SK}}{2^{t}})$$ □

### 6.3. Security Analysis

**Message authentication**: As Theorem 1 states, no polynomial adversary could forge a valid message under the assumption that the ECDLP problem is hard. Therefore, the Central Hospital verifies the validity and integrity of the message $\left({PID}_{{DAU}_{i\;}}{,\;X}_{i}{,\;V}_{i}{,\;t}_{i\;}{,\;u}_{i}^{\prime },\;\sigma_{i}\right)$ by checking whether the equation $d_{i}{P\;=\;y}_{i}+\;\left(X_{i}{\;+\;V}_{i}{\;+\;K}_{pub}W_{4}^{*}\right)H^{*}$ holds or not, where $W_{4}^{*}{\;=\;W}_{4}{(PID}_{{DAU}_{i\;}}{,\;X}_{i\;}{,\;V}_{i})$ and $H^{*}\;=\;H\left({PID}_{{DAU}_{i\;}}{,\;T\;,\;u}_{i}^{\prime }\right)$. Thus, the proposed scheme for MCPS provides message authentication.**Identity privacy protection**: The pseudonym proposed in this paper is divided into two types: the pseudonym of DAUs ${(PID}_{{DAU}_{i\;}},\;1\;\leq \;i\;\leq \;n\;)$ and the pseudonym of patients ${(PID}_{P_{j}},\;1\;\leq \;j\;\leq \;n)$. ${PID}_{{DAU}_{i\;}}$ and ${PID}_{P_{j\;}}$ are generated by combining the randomly chosen secret value $a_{i}$ or $b_{j}$ and the system master key λ. No adversary could compute the real identity from the pseudonym without knowing the secret $a_{i}$ or $b_{i}$ and λ. Thus, the pseudonym proposed in this paper can protect the identity privacy of DAUs and patients.**Resistance to replay attack**: Whenever *DAU_i_* makes an individual signature, it chooses a latest timestamp $t_{i}$. The Central Hospital will check the freshness of the timestamp $t_{i}$ in order to detect the replay attacks.**Resistance to modification attack**: According to Theorem 1, the Central Hospital can protect the integrity of message $\left({PID}_{{DAU}_{i\;}}{,\;X}_{i}{,\;V}_{i}{,\;t}_{i\;}{,\;u}_{i}^{\prime },\;\sigma_{i}\right)$. Therefore, any modification on the message will be detected by checking whether the equation $d_{i}{P\;=\;y}_{i}+\;\left(X_{i}{\;+\;V}_{i}{\;+\;K}_{pub}W_{4}^{*}\right)H^{*}$ holds or not.**Resistance to spam attack [17]**: Because of natural compression property of the aggregate signature, the proposed signature scheme can combine *n* individual signature into one short signature. The length of the aggregate signature will not increase with the increase of the number of signers. Therefore, in the blockchain-based MCPS, more transactions can be added into a block. However, the attacker has to send more transactions to congest the network. It will spend more transaction fee which will increase the cost of spam attacks.

## 7. Efficiency Analysis

Certificateless aggregate signatures can be classified into pairing-based certificateless aggregate signatures and ECC-based certificateless aggregate signatures. In this paper, we adopt the same efficiency evaluation method as reference [11,29], in which the simulations are conducted on an Intel I7 3.4 GHz, 4 GB machine with Windows 7. Pairing-based aggregate signature schemes can be simulated on the bilinear pairing ${e\;:\;G}_{1}{\;\times \;G}_{1}\;\to {\;G}_{2}$. $G_{1}$ is an additive group generated with the order $q_{1}$ on the type A elliptic curve $E_{1}{\;:\;y}^{2}{\;=\;x}^{3}{\;+\;x\;\mathrm{mod}\;p}_{1}$, where ${\;p}_{1}$ and $q_{1}$ are 512-bit and 160-bit prime number, respectively [11]. For ECC-based aggregate signature schemes, the simulation can be conducted over the non-singular elliptic curve ${E\;:\;y}^{2}{\;=\;x}^{3}{\;+\;ax\;+\;b\;\mathrm{mod}\;p}_{2}$. G is an additive group generated on *E* with the order $q_{2}$, where $p_{2}$, $q_{2}$ are two 160-bit prime numbers, respectively. The above mentioned bilinear pairing and elliptic curve constructed in the experiments are on the same security level of 80 bits. As shown in Table 3 and Table 4, the running time of these encryption operations has been presented.

The computation cost and communication cost are two important factors to evaluate certificateless aggregate signature schemes. In this section, the efficiency analysis is divided into two parts. First, we compare the proposed scheme with related certificateless aggregate signature schemes. Second, we compare the proposed scheme with related aggregate signature schemes based on blockchains.

1. The efficiency analysis of certificateless aggregate signature schemes

Table 5 compares the computation cost of the proposed scheme and related certificateless aggregate signature schemes [9,29].
-In the individual sign algorithm, *DAU_i_* needs three scalar multiplications in the elliptic curve and two general hash operations to generate individual signature. The computation cost of our scheme in individual signature is smaller than related certificateless aggregate signature schemes [9,29].-In the individual-verify algorithm, the Central Hospital needs three scalar multiplications, three point addition operations in the elliptic curve, and two general hash operations to verify the *DAU_i_*’s individual signature. The computation cost of our scheme in individual verification is smaller than that of Gong et al.’s scheme [9], but slightly higher than that of Cui et al.’s scheme [29].-As shown in Figure 3, in the aggregate verify algorithm, the Central Hospital needs (2*n*+1) scalar multiplications, (2*n* + 1) point addition operations in the elliptic curve, and 2*n* general hash operations to verify the aggregate signature. The computation cost of our scheme in aggregate verification is smaller than Gong et al.’s scheme [9], but slightly higher than that in Cui et al.’s scheme [29].

Table 6 shows the communication cost of our scheme and related certificateless aggregate signature schemes. In the proposed scheme, the aggregate signature length, such as that of CAS-2 in [9], is a constant, which does not increase with the number of individual signatures.

From Figure 4, we can see that the communication cost of the proposed scheme is obviously smaller than that of CAS-1 [9] and Cui et al.’s scheme [29], and slightly smaller than that of CAS-2 [9].

2. The comparison of certificateless aggregate signatures based on blockchain

In this subsection, we compare the computation cost and communication cost of the proposed scheme with two most recently proposed certificateless aggregate signature schemes based on blockchain [17,18]. As shown in Table 7 and Figure 5, in the individual sign algorithm and aggregate verify algorithm, the computation cost of the proposed scheme is lower than that of Gao et al.’s scheme [18], but it is close to Zhao et al.’s scheme [17]. In the individual verify algorithm, the computation cost of the proposed scheme is lower than Gao et al.’s scheme [18] but slightly higher than that of Zhao et al.’s scheme [17].

As shown in Table 8 and Figure 6, the aggregate signature length of the two most recently proposed certificateless aggregate signature schemes [17,18] based on blockchain is correlated to the individual signature number. However, the aggregate signature length of our scheme is |G|+|q|, which is a constant and is obviously lower than the other two schemes [17,18]. That is to say, the storage capacity of the aggregate signature does not increase with the increase of the *DAU_i_*’s in each transaction, which can effectively improve the storage efficiency of each block.

## 8. Conclusions

In this paper, a certificateless aggregate signature scheme based on blockchain is proposed, which can be used for secure storage and sharing of medical data in MCPS. To improve performance, the function of trapdoor collision calculation in trapdoor hash function is included in our proposed scheme. The security analysis presents that the proposed scheme is existentially unforgeable against adaptive chosen-message attacks, which is resistant to replay attack and modification attack. The proposed scheme provides message authentication and identity privacy protection, which satisfies the security requirements of MCPS. Compared with pairing-based schemes, the scheme proposed in this paper is based on ECC with better computational efficiency, and the computational cost of our scheme is lower. More importantly, the aggregate signature length of the proposed scheme is independent of the number of signers, which can effectively increase the number of transactions stored in each block. Therefore, the proposed scheme can alleviate the capacity limitation of blockchain and prevent spam attacks to a certain extent.

In the future work, we will focus on the lattice-based digital signature algorithm and combine it with blockchain to improve the security of blockchain. More importantly, we will apply our research to practice and obtain measurement results from practical implementation.

## Figures and Tables

**Figure 1 sensors-20-01521-f001:**
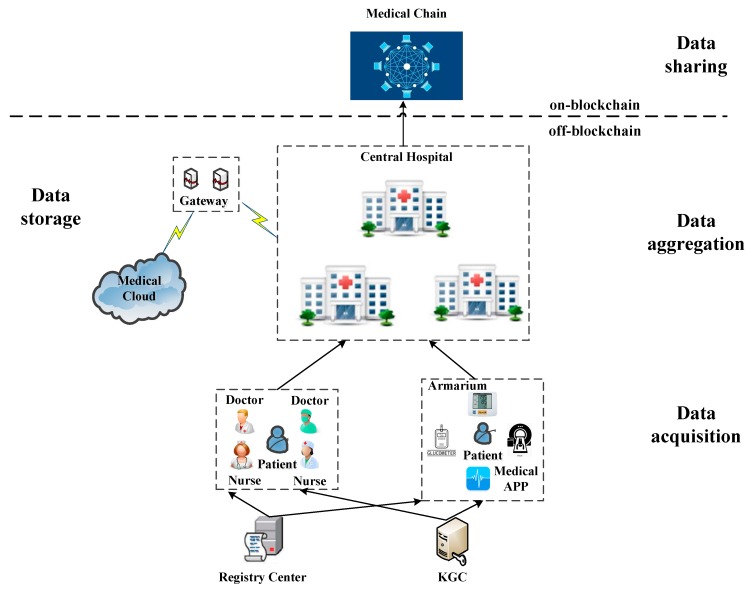
System model.

**Figure 2 sensors-20-01521-f002:**
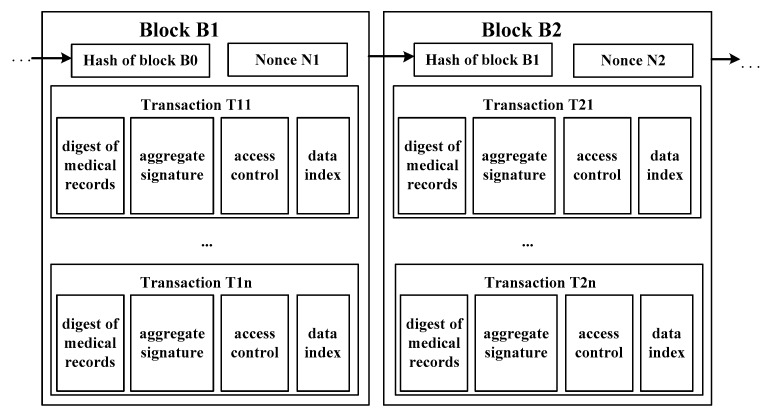
Medical Chain model.

**Figure 3 sensors-20-01521-f003:**
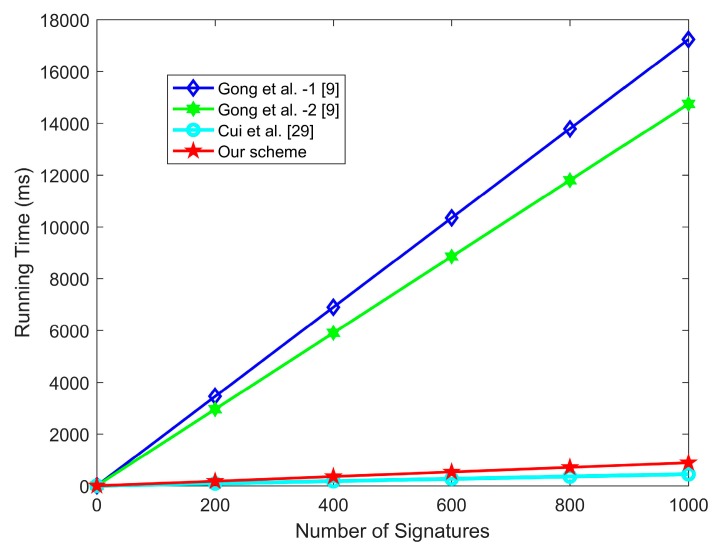
The comparison of aggregate verification time.

**Figure 4 sensors-20-01521-f004:**
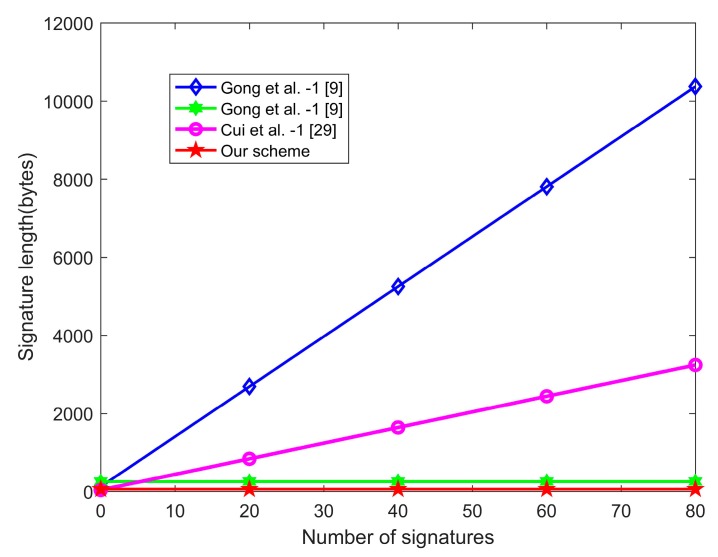
The comparison of signature length.

**Figure 5 sensors-20-01521-f005:**
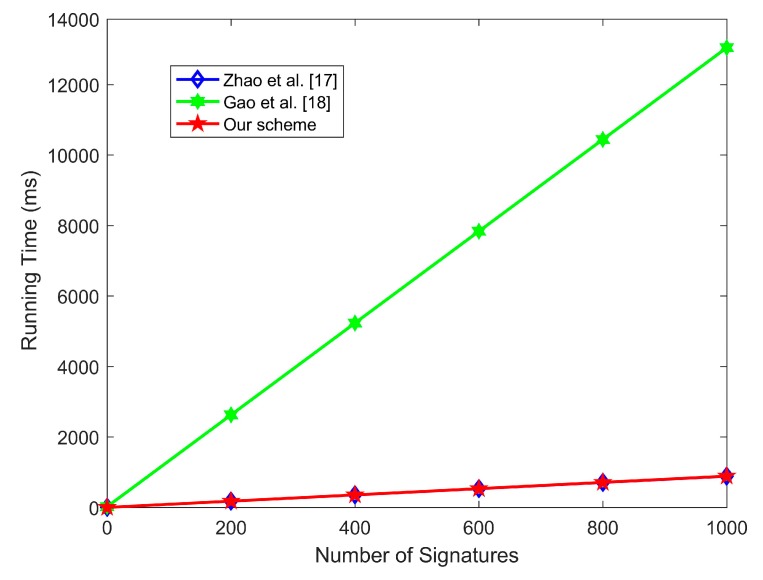
The aggregate verification cost of schemes based on blockchain.

**Figure 6 sensors-20-01521-f006:**
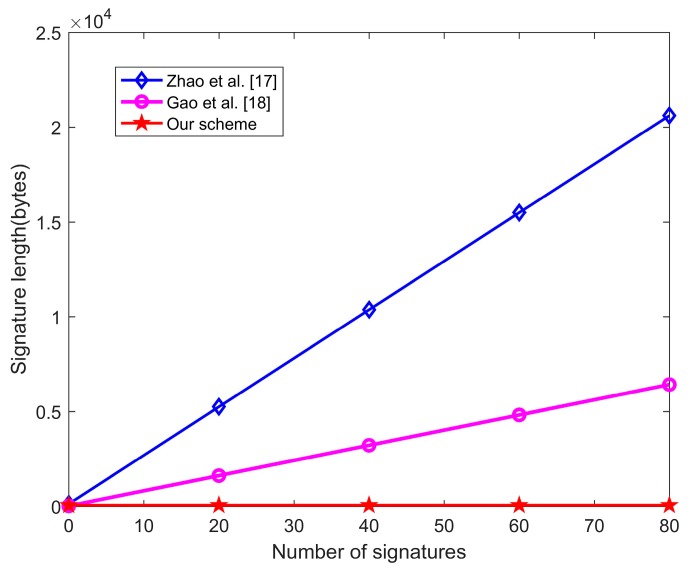
The signature length of schemes based on blockchain.

**Table 1 sensors-20-01521-t001:** The comparison of three types of blockchains.

Blockchain Type	Public Blockchain	Private Blockchain	Consortium Blockchain
Open objects	All	Individuals or inside company	Authorized companies or organizations
Consensus mechanism	PoW, PoS, DPoS	PBFT	PBFT, Raft
Centralization	Decentralization	Centralization	Partial centralization
Typical application	Bitcoin, Ethereum	Overstock	Hyperledger, R3CEV
Characteristics	Self-building of trust	Traceability	Improvement of efficiency

**Table 2 sensors-20-01521-t002:** The comparison of relevant blockchain-based schemes.

Scheme	Integrity	Authentication	Bilinear Maps	Relevance to Number of Users	MCPS	Aggregate Signature
Zhao et al. [17]	Y	Y	N	Y	N	Y
Gao et al. [18]	Y	Y	Y	Y	N	Y
Wang et al. [30]	Y	Y	Y	N	N	Y
Danzi et al. [32]	Y	Y	N	N	N	N
Kaga et al. [33]	N	Y	N	Y	N	N
Our scheme	Y	Y	N	N	Y	Y

**Table 3 sensors-20-01521-t003:** Different encryption operation running time [11,29,37].

Encryption Operation	Description	Time (ms)
$t_{p}$	The bilinear pair operation	4.2110
$t_{mp}$	The scalar multiplication in the bilinear pair	1.7090
$t_{ap}$	The bilinear pair-to-midpoint addition	0.0071
$t_{hp}$	The hash-to-point operation in bilinear pair	4.4060
$t_{mecc}$	The scalar multiplication in elliptic curve	0.4420
$t_{aecc}$	The point addition operation in elliptic curve	0.0018
$t_{h}$	The general hash operation	0.0001

**Table 4 sensors-20-01521-t004:** Group parameter [11,29,37].

Symbol	Description	Length (bytes)
|*G*_1_|	The size of elements in group $G_{1}$	128
|*G*|	The size of elements in group G	40
|*q*|	The size of the elements in $Z_{q}^{*}$	20

**Table 5 sensors-20-01521-t005:** The comparison of computation cost.

Scheme	Individual Sign	Individual Verify	Aggregate Verify
Gong et al. -1 [9]	${2t}_{mp}{\;+\;t}_{ap\;}{+\;t}_{hp}\;$ ≈ 7.8311ms	${3t}_{p}{\;+\;2t}_{hp}\;$ ≈ 21.445ms	$\left(2n\;+\;1\right){\;t}_{p}{\;+\;2nt}_{hp}$ ≈ 17.234n + 4.211ms
Gong et al. -2 [9]	${3t}_{mp}{\;+\;2t}_{ap\;}{+\;2t}_{hp}$ ≈ 13.9532ms	${3t}_{p}{\;+\;t}_{mp}{\;+\;t}_{ap\;}{+3t}_{hp}$ ≈ 27.5671ms	$\left(n\;+\;2\right){\;t}_{p}{\;+\;nt}_{mp}{\;+\;nt}_{ap\;}{\;+\;2nt}_{hp}$ ≈ 14.7391n + 8.422ms
Cui et al. [29]	$t_{mecc}{\;+\;t}_{aecc\;}{+\;t}_{h}$ ≈ 0.4439ms	$3t_{mecc}{\;+\;2t}_{aecc\;}{+\;2t}_{h}\;$ ≈ 1.3298ms	$\left(n\;+\;2\right)t_{mecc}{\;+\;2nt}_{aecc}{\;+\;2nt}_{h\;}$ ≈ 0.4458n + 0.884 ms
Our scheme	$t_{mecc}{\;+\;3t}_{h\;}$ ≈ 0.4423ms	$3t_{mecc}+\;3t_{aecc}{+\;2t}_{h\;}$ ≈ 1.3316ms	$\left(2n\;+\;1\right)t_{mecc}\;+\;\left(2n\;+\;1\right)t_{aecc}{\;+\;2nT}_{H\;}$ ≈ 0.8878n + 0.4438ms

**Table 6 sensors-20-01521-t006:** The comparison of communication cost.

Scheme	Aggregate Signature Length	Correlation between Signature Length and *n*
Gong et al. −1 [9]	${(n\;+\;1)|G}_{1}|$	Yes
Gong et al. −2 [9]	${2|G}_{1}|$	No
Cui et al. [29]	(n + 1)|G|	Yes
Our scheme	|G| + |q|	No

**Table 7 sensors-20-01521-t007:** Computation cost of schemes based on blockchain.

Scheme	Individual Sign	Individual Verify	Aggregate Verify
Zhao et al. [17]	$t_{mecc}{\;+2t}_{h}\;$ ≈ 0.4422ms	${2t}_{mecc}{\;+2t}_{h}$ ≈ 0.8842ms	$\left(2n\;+\;1\right){\;t}_{mecc}{\;+\;2nt}_{h}$ ≈ 0.8842n + 0.442ms
Gao et al. [18]	${5t}_{mp}{\;+\;3t}_{ap\;}{+2t}_{hp}$ ≈ 17.3783ms	$5t_{p}{\;+\;3t}_{hp}\;$ ≈ 34.273ms	$\left(n\;+\;4\right)t_{p}{\;+\;(2n+1)t}_{hp}$ ≈ 13.023n + 21.25 ms
Our scheme	$t_{mecc}{\;+\;3t}_{h\;}$ ≈ 0.4423ms	$3t_{mecc}+\;3t_{aecc}{+\;2t}_{h\;}$ ≈ 1.3316ms	$\left(2n\;+\;1\right)t_{mecc}\;+\;\left(2n\;+\;1\right)t_{aecc}{\;+\;2nT}_{h\;}$ ≈ 0.8878n + 0.4438ms

**Table 8 sensors-20-01521-t008:** Communication cost of schemes based on blockchain.

Scheme	Aggregate Signature Length	Correlation between Signature Length and *n*
Zhao et al. [17]	2n |G|+|q|	Yes
Gao et al. [18]	${(2n\;+\;1)|G}_{1}|$	Yes
Our scheme	|G|+|q|	No

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
