# Peer review of "An Efficient Certificateless Aggregate Signature Scheme for Blockchain-Based Medical Cyber Physical Systems"

_sensors, 2020, doi:10.3390/s20051521_

Round 1

Reviewer 1 Report

This paper proposes a certificateless aggregate signature scheme for blockchain-based MCPS. In general, the problem to be solved is well described, and the theoretical solution and the comparison with previous studies are well described.
Nevertheless, I would like to ask the author a question and a corresponding request.
Trapdoors were used to improve the performance, but the impact of collisions seems to be insufficient. Is there any impact on the accuracy of the collision? I think it would be a better paper if analysis was added.

Reviewer 2 Report

In this paper, the authors propose a certificate-less aggregate signature scheme based on multi-trapdoor hash function which exploits its natural compression properties to improve the efficiency of blockchain-based medical cyber physical systems and reduce the computation cost. Further, in order to address the capacity limitations of blockchain performance, the authors propose a data storage and sharing model where medical data is stored off-blockchain and shared on-blockchain. An efficiency analysis conducted by the authors reveals the important computation-communication cost tradeoff of the proposed scheme as well as the comparative performance with respect to benchmark approaches. An important take-away message of the paper is that the aggregate signature length of the proposed scheme is independent of the number of signers, which can increase the number of stored transactions in each block.

Comments

- When stating the contributions of the paper, the third bullet (Page 2, line 78) does not correspond to an actual contribution but rather it elaborates on the properties of the proposed scheme. Instead, the authors should rephrase the summary of their contributions and reflect only on the novelties they propose beyond the state of the art approaches.

- Please motivate why the specific security model is adopted in Section 3.3.2.

- In Section 3.5, are all those listed security requirements equally important for the medical cyber-physical systems? Can any of them be prioritized over some others? How these requirements for medical data security relate with the choice of an attribute-based signature?

- For clarity, it would help if the interactions of the proposed certificate-less aggregate signature scheme described in Section 4, could be directly associated with Figure 1 (System Model).

- It would add value if the authors commented on the vulnerability of their proposed scheme against other types/models of security attacks.

Other comments/Typos

Some typos were spotted and the authors are highly encouraged to perform a new proofreading round over the entire text. Examples:

Abstract, Line 17: “rely”

Page 2, Line 65: “increase”

Page 2, Line 74: “two-layer”

Page 4, Line 147: “could not”

Page 5, Line 204: “system”

Page 5, Line 213: “certificate-less”

Reviewer 3 Report

The authors present a certificateless aggregate signature scheme based on blockchain, which can be used for secure storage and sharing of medical data in healthcare CPS. 

The topic is novel and important, the paper tends towards theoretical. The authors should go beyond simulations and provide measurement results from a practical implementation. 

The MCPS concept should be described in more detail, especially how cyber and physical components could be securely co-simulated.

Also, different types of blockchains should be evaluated. Moreover, quantum safe encryption should be considered.

In the conclusion the authors should envision future work.

The references should be more recent (i.e. 2018-2019) and include the following:

- Chen, Yi, Shuai Ding, Zheng Xu, Handong Zheng, and Shanlin Yang. "Blockchain-based medical records secure storage and medical service framework." Journal of medical systems 43, no. 1 (2019): 5.
- Suciu, George, Carmen Nădrag, Cristiana Istrate, Alexandru Vulpe, Maria-Cristina Ditu, and Oana Subea. "Comparative analysis of distributed ledger technologies." In 2018 Global Wireless Summit (GWS), pp. 370-373. IEEE, 2018.
- Sengupta, Jayasree, Sushmita Ruj, and Sipra Das Bit. "A comprehensive survey on attacks, security issues and blockchain solutions for IoT and IIoT." Journal of Network and Computer Applications (2019): 102481.

Reviewer 4 Report

This paper proposes a certificateless aggregate signature scheme for blockchain-based MCPS that supports decentralization, security, credibility and tamper-proof, which is suitable for solving the problem of authentication of related medical staff, medical  equipment and medical APPs in MCPS.

The paper has several issues that need to be addressed:

The writing style needs to be improved. The abstract is not well written because the contribution is just discussed in the second sentence. The authors should specify the motivation and the novelty of their scheme. If capacity limitation and bilinear map-based constructions are the main problems, we have several articles in IoT that constructed secure schemes without bilinear maps, then specify why these schemes cannot be adopted. Therefore the abstract needs to be properly reformulated. In the same way,  looking at the introduction, we do suggest that the authors read other journal papers to see how to improve the introduction. As far as Iknow, the introduction should have a clear motivation and needs to cite a few relevant references and show why those references cannot directly be adopted for the scheme. In related work, the authors are comparing their work with reference 24 and 28 for performance and references 8,13,16 for MCPS related schemes. Surprisingly, nothing is said for the 8, 13,16. Their drawbacks nor their contributions. It seems like the authors did not fully consider those fey references I do assume that the main target is to propose a scheme for MCSP scenario. Thus as it is done in other many works, at least a table showing which security primitives does your scheme achieve compared to existing relevant work is necessary. The references 8,13,16 according to me are not very suitable for MCSP because those schemes so not address deeply  MCPS scenario. There are schemes that use blockchain to offer integrity. Therefore the authors should find suitable references for MCPS or they can remain with their first contribution which is based on signature length

Round 2

Reviewer 1 Report

I was able to review an interesting study.

I hope you have a good result.

Author Response

Dear reviewer:

We would like to thank you for your valuable and enriching comments, suggestions. They are helpful for us to improve our writing skills. 

Thank you and best regards.

Yours sincerely,

Hong Shu

Reviewer 2 Report

The authors have provided a revised version of the manuscript, addressing most of the reviewers' comments.

When discussing the novelty of their approach, the authors state: “Although these blockchain-based schemes can provide data integrity for IoT applications, they are not suitable for our MCPS application because they are not aggregate signature schemes.” I believe the authors should motivate the choice of their proposed method in a different way, emphasizing on the specific advantages it brings compared to other blockchain-based schemes. By simply stating that literature approaches are not suitable for MCPS applications without mentioning convincing arguments, their contribution on this research gap seems limited, especially when certificate-less aggregate signatures have not been widely used in MCPS.

Further proofreading - especially over the newly added text - is required.

Reviewer 4 Report

The comments were addressed accordingly

Author Response

Dear reviewer:

We would like to thank you for your valuable and enriching comments, suggestions. They are very helpful for us to improve our writing skills. 

Thank you and best regards.

Yours sincerely,

Hong Shu